# **Evaporation from Cultivated and Semi-Wild Sudanian Savanna in** West Africa

Natalie C. Ceperley<sup>1,2</sup>, Theophile Mande<sup>2</sup>, Nick van de Giesen<sup>3</sup>, Scott Tyler<sup>4</sup>, Hamma Yacouba<sup>5</sup>, Marc B. Parlange<sup>1,2</sup>

<sup>5</sup> <sup>1</sup>Department of Civil Engineering, Faculty of Applied Sciences, University of British Columbia, Vancouver, British Columbia, V6T 1Z4, Canada

<sup>2</sup>Laboratory of Environmental Fluid Mechanics and Hydrology, School of Architecture, Civil and Environmental Engineering, Swiss Federal Institute of Technology, Lausanne, 1015, Switzerland

3Department of Civil Engineering and Geosciences, Delft University of Technology, 2600 AA, Delft, Netherlands

<sup>4</sup>Department of Geological Sciences & Engineering, University of Nevada, Reno, Nevada, 89557, United States of America <sup>5</sup>Laboratory Hydrology and Resources in Water, International Institute for Water and Environmental Engineering (2iE), Ouagadougou, 01, Burkina Faso

Correspondence to: Natalie C. Ceperley (natalie.ceperley@unil.ch)

- Abstract. Rain-fed farming is the primary livelihood of semi-arid West Africa. Changes in land cover have the potential to affect precipitation, the critical resource for production. Turbulent flux measurements from two eddy-covariance towers and additional observations from a dense network of small, wireless meteorological stations combine to relate land cover (savanna forest and agriculture) to evaporation in a small (3.5 km<sup>2</sup>) catchment in Burkina Faso, West Africa. We observe larger sensible and latent heat fluxes over the savanna-forest in the headwater area relative to the agricultural section of the
- watershed all year. Higher fluxes above the savanna-forest are attributed to the greater number of exposed rocks and trees and the higher productivity of the forest compared to rainfed, hand-farmed agricultural fields. Vegetation cover and soil moisture are found to be primary controls of the evaporative fraction. Satellite derived vegetation index (NDVI) and soil moisture are determined to be good predictors of evaporative fraction, as indicators of the physical basis of evaporation. Our measurements provide an estimator that can be used to derive evaporative fraction when only NDVI is available. Such
- large-scale estimates of evaporative fraction from remotely sensed data are valuable where ground-based measurements are lacking, which is the case across the African continent and many other semi-arid areas. Evaporative fraction estimates can be combined, for example, with sensible heat from measurements of temperature variance, to provide an estimate of evaporation when only minimal meteorological measurements are available in remote regions of the world. These findings reinforce local cultural beliefs of the importance of forest fragments for climate regulation and may provide support to local
- decision makers and rural farmers in the maintenance of the forest areas.

#### **1** Introduction

conditions (Foken, 2008; Szilagyi and Parlange, 1999).

The sudanian savanna in South-Eastern Burkina Faso is a patchwork of savanna, forest, and scrubland with some patches more representative of the drier Sahel, and others more representative of the more humid Guinean forests. Vegetation is mainly deciduous according to seasonal moisture availability, but spatial variations in topography, water availability, and

- plant communities result in some variation in greenness and some evergreen species, for example near the springs. Historically, people in this region rely on a mix of hunting and gathering, small scale agriculture, and pastoralism. As land claims and regulations have changed, communities have been forced to rely more on agricultural production as a primary source of food and income, resulting in land conversion for agriculture. Today small-scale rain fed agriculture is the dominant livelihood in large parts of West Africa, despite its high level of dependence on seasonally controlled hydrology.
- Conversion of the landscape to agriculture involves removing rocks, trees, natural grasses, and tilling the soil (Swanson, 1978). Transformation of forestland to agriculture has been shown by model simulations to alter global circulation, hydrology, and biogeochemistry both in the present and in predictions of the future (Abiodun et al., 2008; Feddema et al., 2005; Mande et al., 2011; Steiner et al., 2009; Sylla et al., 2015; Vitousek, 1997).

Modification of the land surface results in changes to the physical environment, and specifically hydrological fluxes, by altering the components of the surface energy budget (Pielke et al., 2002). The partition of net radiation into sensible heat, evaporation, and soil heat flux drives global atmospheric processes and is controlled by interacting surface and atmospheric

The energy balance is challenging to close even in areas and regions with extensive datasets and accessibility, and is particularly challenging in areas with complex surfaces (Burba, 2013; Domingo et al., 2011; Farhadi, 2012; Federer et al.,

- 2003; Foken et al., 2009; Guo et al., 2006; Katul and Parlange, 1992; Krishnan et al., 2012, 2012; Kustas et al., 1994; Parlange and Katul, 1992; Williams et al., 2012). Evaporation from vegetated surfaces remains the component of the global distribution of water that is the least frequently measured and thus the least well understood (Brutsaert, 1982; Brutsaert and Parlange, 1992; Burba, 2005; Compaore, 2006; Crago, 1996; Crago and Qualls, 2013), particularly in West Africa, due to limited field observations (Bagayoko et al., 2007; Dolman et al., 1997; Gash et al., 1997; Mande et al., 2011). A connection
- exists between changes in albedo and the occurrence of drought in West Africa, although the physical processes and direct implications for desertification are debatable (Charney, 1975; Nicholson et al., 1998). The evaporative fraction, the ratio of latent heat flux to available energy, is useful to estimate total daily evaporation with measurements of a single component of the energy balance and to up-scale surface measurements using remote sensing products (Brutsaert and Sugita, 1992; Compaore, 2006; Porte-Agel et al., 2000; Shuttleworth, 1989; Szilagyi et al., 1998;
- Szilagyi and Parlange, 1999). Using evaporative fraction to calculate the total daily evaporation is based on the concept of self-preservation in the diurnal evolution of the surface energy budget (Brutsaert and Sugita, 1992; Porte-Agel et al., 2000), stating that the diurnal cycle of each of the energetic fluxes will resemble that of available energy, even if there is variation in the quantity, allowing for exploiting satellite data that are typically only obtainable once a day at best. Remotely sensed

land surface temperature is currently the primary tool for mapping the surface energy budget over a large area (Bateni and Entekhabi, 2012). Evaporative fraction is constant during day time in fair weather conditions (Gentine et al., 2007) but can be much less constant when moisture circulation rates are high and available soil moisture increases (Lhomme and Elguero, 1999). Seasonal progression of evaporative fraction response to rainfall and moisture availability can depend on surface

5 conditions. For example, it can respond faster in grassland than in woodland (Farah et al., 2004). These variations are not explained by meteorological conditions, including cloudiness alone, but rather change in surface resistance and moisture advection and availability (Farah et al., 2004; Lohou et al., 2010, 2014).

We measured the complete energetic and hydrologic fluxes in two sites of a semi-arid, mixed-use catchment over a year and a half, capturing both the greening and dry-down phases. The land is used as a agroforestry parkland farmed every 2-3 years and a forested area made up of evergreen trees arranged in a gallery forest surrounding a spring and an open wooded savanna (savanna-forest) on a plateau about 100 meters above the surrounding land. These land covers are representative of the surrounding region and capture the range from more to less anthropogenic land uses. The multi-use comparison over multiple seasonal cycles puts this study among the few recent, long-term studies in this region (Bagayoko et al., 2007; Brümmer et al., 2008; Ezzahar et al., 2009; Guichard et al., 2009; Guyot et al., 2009, 2012, Lohou et al., 2010, 2014,

- Mamadou et al., 2014, 2016; Mauder et al., 2006; Ramier et al., 2009; Timouk et al., 2009; Velluet et al., 2014). We calculate the evaporative fraction over the study period and compare it with land cover and atmospheric controls, in order to provide estimation based on the physical basis of fluxes. Our measurements are significant because they allow calibration and comparison for calculation of the components of the energy budget from lower cost and more easily maintained stations (Nadeau et al., 2009; Simoni et al., 2011) and corresponding data from satellite imagery. This study has implications for
- development priorities, as it takes place in context where local livelihood can be dramatically affected by slight changes in the water balance and land cover.

#### 2 Measurements and Calculations

# 2.1 Site Description

Observations were made in a small catchment (3.5 km<sup>2</sup> area) neighboring the village of Tambarga in the commune of Madjoari, in the Gourma Province, in Burkina Faso, West Africa (figure 1). The ephemeral stream defining the catchment (outlet is 11°26'29.7"N 1°12'57.7"E) flows into the Singou River, which joins the Pendjari River and eventually flows into the White Volta of the Volta River Basin, the third largest river basin by area in West Africa, after the Niger and the Senegal. A rocky escarpment defines the catchment with a plateau on average some 100 meters above the lower agricultural fields and the soil is predominantly sandy-loam (Ceperley, 2014). These fields are the "house" farms and are smaller than the main

revenue farms. In 2009, they were farmed with short rotation millet and in 2010 they were left fallow. Plowing occasionally uses animal-drawn plows, but is primarily done by hand – this is not intensive agriculture. The open wooded savanna (savanna-forest) on top of the rocky escarpment and rain-fed grain (corn or millet) are the two dominant land covers of the

catchment according to area. At opposite ends of the catchment, there is a dense gallery forest in the valley that grows near perennial springs and an ephemeral wetland used for rice cultivation near the point considered the outlet of the watershed. The existence of a raised plateau with perennial springs suggests that there may be lateral subsurface water transfer. Farming is the main livelihood in the village and crops include millet, sorghum, cotton, and rice. Agroforestry trees in the fields are

- common and consist most often of the tree species Vitellaria paradoxa, Sclerocarya birrea, and Ficus sp. (Bordes, 2010, see table 2). Burning mostly occurs between November and January, but occasionally there is a fire in February or March. The village is made up of a majority of people from the Gourmantche ethnic group, though there is a significant population of Peulh, and some migrants of other of Burkina Faso neighboring areas or countries (Ceperley, 2014).
- The watershed falls in the sudanian zone of the West African monsoon climate system, defined by alternating wet and dry seasons, with the rain falling between May and September. The natural vegetation is sudanian-wooded savanna, composed of a mix of deciduous woody trees, shrubs, and tall grasses. In addition, due to the variation in topography and water availability, there are gallery forests near streams or rivers that contain many species endemic further south, in the Guinean zone. The surrounding area is a patchwork of hunting reserves and national parks, and thus has a higher level of vegetation
- cover than most of the country. This watershed is a prime location to study the consequences of land use change from sudanian savanna to agricultural fields, since it contains both open wooded savanna that hasn't been memorably farmed and regularly farmed fields. Agriculture is primarily rainfed and not mechanized corn and millet cultivation. In addition, the surrounding sudanian savanna, which is characterized by fire-selected grasses ranging from 20 centimeters to 1.5 meters in height also includes patches of woody scrub-land, open forests, gallery forests, and riparian stands (Arbonnier, 2004).
- Inventory of woody species taller than breast height in the two major land covers was inventoried according to Adamou (2005) and is reported in Table 2.

#### **2.2 Field Measurements**

Two energy balance stations were installed from May 2009 through September 2010. One was situated in an agricultural field planted with short season millet in 2009 and left fallow in 2010, and the second one measured over the gallery forest

- when the wind came from the West (90°±45°) and over the open wooded savanna when the wind came from the South (180°±45°). They were equipped with sonic anemometers, infrared open path gas analyzers, net radiometers, and air temperature and humidity sensors (table 1). Eddy-covariance equipment was placed facing two opposite directions (46° and 226°) on the lower station over the field and in the dominant wind direction on the upper station over the gallery forest. The distance between the two measurement points was approximately 1 kilometer and 100 meters of height difference. Nearby
- the station in the field was a high precision rain gauge measuring at a resolution of 0.1 mm. In addition, a network of up to 12 small meteorological stations (Ingelrest et al., 2010), was distributed across the watershed with sensors to measure incoming solar radiation, wind direction and speed, air temperature and relative humidity, rainfall, soil moisture and temperature, and surface temperature. In this analysis, we use data from May 2009 - October 2010, the 15 months with both
  - 4

towers operational, but when possible, we present the longest time series possible for climatic context. We attempted to measure ground heat flux using heat flux plates but ultimately rejected the observations because of irregularities in the land surrounding the plates.

### 2.3 Flux Calculation

The surface energy budget is written in Eq. (1):

$$R_n = L_e E + H + G_{,\,}(1)$$

where  $L_eE$  is latent heat flux, *H* is sensible heat flux,  $R_n$  is the net radiation, and *G* is the soil heat flux, all in Watts per square meter (W m<sup>-2</sup>).

The sensible heat, is expressed in Eq. (2):

$$H = \rho c_n \overline{w'T'}$$
, (2)

where  $\rho$  is the air density (kg m<sup>-3</sup>),  $c_p$  is the specific heat (J kg<sup>-1</sup> K<sup>-1</sup>), w'T' is the covariance of fluctuations of vertical wind speed (m s<sup>-1</sup>) and temperature (K). Latent heat flux, is expressed in Eq. (3):

$$L_e = L_e \rho \overline{w'q'}, (3)$$

 $L_e$  is the latent energy of vaporization (J g<sup>-1</sup>), and w'q' is covariance of fluctuations of vertical wind speed (m s<sup>-1</sup>) and 15 humidity (g m<sup>-3</sup>).

All measurements were taking at 10 Hz, and fluxes of sensible (H) and latent ( $L_eE$ ) heat were calculated at a half hour time step using the covariance calculations as written above. Only day light measurements, consistently between sunrise and sunset (8 am - 4 pm), corresponding to when energetic fluxes were of significant magnitude, were used for the comparison. The total of turbulent fluxes, H +  $L_eE$ , was subtracted from the net radiation,  $R_n$ , to give an indicator of ground heat flux (G,

- see equation 1), any unaccounted for flux transfers, and the error (Brutsaert, 1982; Higgins, 2012). We used a planar fit correction that effectively tilted measurements of the three components of the wind field perpendicular to the direction of flow, so that the vertical wind was equal to zero over one month averaging periods (Aubinet et al., 2012; Burba, 2005; Oldroyd et al., 2015; Rebmann et al., 2012; Wilczak et al., 2001). We then performed a linear regression using the mean wind vectors to obtain a matrix that we used to adjust wind vectors and stress tensors in a new coordinate system
- with a z-axis perpendicular to the mean streamline. Finally, we rotated the intermediate winds and stress tensors. The Webb-Pearman-Leuning equation (Foken et al., 2012; Leuning, 2007; Webb et al., 1980) Eq. (4):

$$E = (1 + \mu\sigma) \left( \overline{\omega' \rho'_{v}} \right) + \frac{\overline{T}}{\overline{\rho_{v}}} \overline{\omega' T'}.$$
(4)

was used to correct for any influence of trace gas concentrations on temperature and humidity fluctuations.

# **2.4 Evaporative Fraction**

Evaporative fraction (EF) was calculated for each half hour of data, separately over the savanna-forest and the agricultural area by dividing the latent heat flux by the available energy, which is equivalent to the sum of sensible and latent heat flux. Although the true measure of available energy would be the difference between the net radiation and the ground heat flux,

the sum of the turbulent fluxes, H and LE, was deemed more accurate given the rejection of our ground heat flux measurement. The midday average (10 am - 2 pm) was used as the EF for a given day as that is when it was the most stable over the year (figure 3) Eq. (5):

$$Ef = \frac{L_e E}{H + L_e E} \,. \, (5)$$

#### 2.5 Volumetric Water Content

- Volumetric water content (VWC, m<sup>3</sup> m<sup>-3</sup>) in the soil was monitored at 15 and 30 cm depths in 2009 and at 5 and 20 cm depths thereafter at some of the small meteorological stations representative of the various land covers (Table 1) using a measure of soil dielectric permittivity and converted to VWC (Topp et al., 1980). Measurements were averaged on the half hour time step for comparison with EC measurements and by day for comparison with EF. Gaps in measurement were due to sensor malfunction. A vertical, spatial average of measurements at various depths was used to obtain a continuous record for the watershed
- the watershed.

# 2.6 Cloud Cover

Cloud cover was calculated by dividing the incoming shortwave radiation (W  $m^{-2}$ ) measured with a radiometer (table 1) at each small meteorological station by the theoretical incoming radiation (W  $m^{-2}$ ) calculated with a simple model (Whiteman and Allwine, 1986) for each of the small meteorological stations operating on any given day Eq. (6):

$$CC = \frac{sw_{measured}}{sw_{Whitman&Allwine}}$$
. (6)

The cloud cover was calculated independently for all stations and then averaged to give a single value per day.

### 2.7 Vegetation Index

Normalized difference vegetation index (NDVI) is based on the amount of infrared radiation absorbed, which is related to the amount of photosynthesis taking place. It is considered a measure of the density of chlorophyll. NDVI is a ratio of the near infrared (NIR) to red wavelengths Eq. (7):

$$NDVI = \frac{NIR-Red}{NIR+Red}$$
 (7)

Seasonal change in NDVI was observed by extraction of the area of interest from the 250-meter resolution West Africa eMODIS 10-day temporally smoothed data (USGS FEWS NET). It has been corrected for molecular scattering, ozone absorption, and aerosols and then smoothed using a least square linear regression (Swets et al., 1999). The NDVI values are

validated with *in situ* observation and photographs. The pixels that contained our stations were extracted to give a catchment – wide seasonal impression of the vegetation change. All of the pixels that cover our catchment are composed of multiple vegetation types, given the relatively coarse resolution, but they have been sorted by dominant land cover. An inventory of woody species (diameter at breast height, 1.3 m, greater than 10 cm), was performed for 1 hectare of the forest – savanna and

5 the entirety of the agricultural area within the catchment (table 2). Vegetation was classified according chorology and life form (Adomou, 2005).

# 2.8 Wind Sector Partition

Dominant wind direction for each half hour covariance measurement was used to sort the sensible and latent energy fluxes (Figure 6). Computation of the mean flux according to wind direction allowed for examination of the effect of wind direction, and corresponding land surface, on the flux magnitudes.

# **3** Results

#### 3.1 Seasonality

A total of 1600 millimeters (mm) of precipitation was measured over the period of intensive monitoring, 2009-2010, 789 mm in 2009 and 811 mm in 2010 and it fell almost entirely during the period from May to October. As seen in figure 2, average monthly air temperature, cloud cover, soil moisture and NDVI followed the seasonal cycle of the rain: temperature was higher in the dry season (November - March) than in the wet season (May - October); cloud cover was lower in the dry

- was higher in the dry season (November March) than in the wet season (May October); cloud cover was lower in the dry season and increased starting in March and April peaking both years in July; soil moisture was highest in the wet season; and vegetation, as shown by NDVI, increased over the course of the wet season, starting in May and peaking in September and declining afterwards as grasses senesced. The high level of seasonality is characteristic of semi arid environments.
- Separation between the lower and upper parts of the catchment is apparent in the NDVI time series, where the savanna-forest consistently stays more green, with the field only surpassing it due to its delayed senescence in September 2010. Plowing, early season harvests, and late season harvests are visible earlier in some years over only the agricultural land. However, since these are averages over a few potentially mixed pixels, which were composed of multiple crops, the individual behaviors are not visible. The land use differentiation is visible even at 250 m pixel resolution.

#### 25 **3.2** Components of Energy Balance

#### 3.2.1 By day

The average diurnal cycle of the energy balance varied according to diurnal cycles and by month (figure 3). In this figure, we compare a single day in April with a single day in July. April is before the wet season begins, and prior to any vegetation growth in the agricultural field, however, some of the evergreen forest canopy is already visible in the photos. By July, crops

or fallow are growing in the field and the canopy is greener. This change in vegetation cover and moisture availability is apparent in the diurnal patterns of the energy balance for both sites. Net radiation is slightly higher during the wet season for both land covers, and none of the fluxes are as smooth, which can be explained by the presence of atmospheric humidity and cloud cover. The sensible heat is higher for both land covers in the dry season, however over the savanna-forest there is still

- 5 latent flux that nearly matches the sensible heat flux even in the dry season. By July, the latent heat flux surpassed the sensible heat flux for both land covers. This can be explained by moisture availability, as the soil moisture content is much higher in both cases. Over the savanna-forest, we can see that the latent heat flux doesn't decline until the late afternoon, suggesting that it is radiation limited and not moisture limited, whereas over the field it peaks closer to midday. The residual, which is a combination of the ground heat flux and any error, is lower over the savanna-forest, and is negative in the
- 10 afternoon. For the month of April, the noon median residual is 20% of the noon net radiation for the agricultural field and 13% for the savanna forest and for the month of July, it is 31% for the agricultural field and 25% over the savanna forest. The evaporative fraction is correspondingly higher in the rainy season than in the dry season and as we would expect, has a higher value over the savanna-forest in the dry season but over the agricultural field in the wet season.
- Net radiation was the most similar component between the two sites, although during the "dust" season of March and April, 15 it was lower over the agricultural field. Sensible heat was greater over the savanna-forest for all months, with the greatest difference in the "hot" period of March through May, which is an important period for the triggering of early convective storms.

There is a scale discrepancy between the eddy covariance measurements and the net radiometer measurements since the latter only senses exchanges directly above and below it whereas the former's range of detection can span a larger area depending on the wind speed. To account for this, we modeled the net radiation at each small station and then compared it to

- that measured with net radiometers with acceptable results (see supplementary material). Latent heat flux was also observed to be greater over the savanna-forest compared to the agricultural field. The point in the day when the latent heat flux peaks signals the moment when the system becomes moisture limited; during the early part of the year, the dry season through May, the diurnal cycle of latent heat flux peaked over the agricultural field during the mid
- 25 morning, from 9 to 10 am, whereas over the savanna-forest, during the same period, the peak in the diurnal was after noon. In general, our data shows that latent heat flux was greatest at early in the diurnal cycle during the dry season, which suggests depletion of all available moisture early in the day.

# **3.2.2** Over entire study period

20

Time series of turbulent fluxes, soil moisture, normalized difference vegetation index, and rainfall demonstrate the highly 30 seasonal moisture and energy availability (figure 4). A high correlation exists for net radiation and the sum of turbulent fluxes (figure 5) between the savanna-forest and the field, with more scatter occurring when soil was wetter (blue). However, the sum of the turbulent fluxes is higher over the savanna-forest than over the field. Since there are equal amounts of net radiation, we can deduce that there is a greater ground heat flux in the field. The lack of shading in the field, and the greater abundance of trees, with a high level of productivity, and rocks support this observation. Examination of the two components of net radiation - net long wave and net short wave - shows that soil moisture exerts much greater control on net long wave radiation, with the change in net long wave radiation according to changes in soil moisture in the field much more apparent. Although there is more scatter in net shortwave when soil is wetter, it is less uniformly a response to the two land

- 5 covers. The savanna-forest's net long wave radiation is greater when the soil is dry whereas the agricultural area has greater net long wave when it is wet. Sensible heat over all land covers is greater under dry conditions than wet (blue), but both sensible and latent heat fluxes are greater over the savanna-forest than the agricultural field regardless of soil moisture. Furthermore, in figure 6, we see that each wind sector has a distinct signature of when latent heat flux is greater than sensible
- heat flux. This variation can be explained because certain features, such as the ephemeral wetland and the gallery forest
  (shown in the top left of the forest plot, 16- 46 degrees), contributed to higher fluxes that have access to moisture that persists longer into the dry season. Over the agricultural field, there is more scatter whereas over the savanna-forest, there is a minimal level of about -170 W/m<sup>2</sup> for net long-wave radiation. In this case, the tree canopy buffers the bare ground from the radiation loss.

#### 3.2.3 Month by month

- Figure 7 shows the two contrasting trends in surface heat fluxes by comparison of the month-by-month ratios between the measurements over the savanna-forest and agricultural land. The savanna-forest contributed more sensible and latent heat flux throughout the year (ratio < 1), but the difference in sensible heat flux between the two sites was greatest at the end of the year, the beginning of the dry season, and the difference latent heat flux between the two sites was greatest at the beginning of the year, after land was cleared by burning. These trends were consistent over the two years of measurement,</p>
- however there were some months (July and August 2010) when the sensible heat was close to equal in the two sites. The higher level of similarity in sensible heat between the two sites in 2009 can be explained by the crop choice that year; That field was planted with early (60-95 day) maturing pearl millet crop compared to its usual late variety (130-150 days), requiring a unusually late tilling and an unusually early harvesting, resulting in bare ground during the growing season. These differences are also visible in the NDVI (figures 2, 4). Net radiation was more similar than the other fluxes; the
- greatest difference occurred when there is bare ground in the field, at end of the dry season, suggesting a higher albedo during this time.

#### **3.3 Evaporative Fraction**

#### 3.3.1 Correlations with surface and atmospheric conditions

Examination of the relationship between evaporation and the environmental variables that dominate in various models, 30 shows that for our site, soil moisture and vegetation cover have the strongest positive correlation with evaporative fraction (figure 8). Over both the savanna-forest and the field, we see that landscape moisture availability, expressed as both NDVI and soil moisture (VWC), exert a strong influence on the evaporative fraction, with higher rates of evaporation occurring at higher levels of soil moisture and vegetation cover, or in other words moisture availability from either plant or soil. Total net radiation does not show a strong influence, suggesting that this is not a radiation-limited system.

Wind speed shows a strong negative correlation with more evaporation occurring at lower wind speeds, contrary to standard 5 evaporation models. Evaporative fraction and the cloud cover exhibit a positive correlation both over the field and the savanna-forest and could be explained by a two part discontinuous function, with a break at 0.4 (Brutsaert, 1982). In a radiation limited system, cloud cover would reduce evaporative fraction, but in this case, since it is positive, we can deduce that cloud cover is an indicator of high rates of evaporation and moisture availability, thus further supporting our hypothesis that this is a moisture limited system.

#### 10 3.3.2 Explanatory model

The relationship between soil moisture, vegetation index, and evaporative fraction can be fitted with a linear regression (figure 9), Eq. (8):

 $EF_{agriculture} = 0.41 \cdot NDVI + 1.4 \cdot VWC + 0.27$  $\{EF_{Forest-Savana} = 0.48 \cdot NDVI + 0.35 \cdot VWC + 0.34^{(8)}\}$ 

Evaporative fraction depends on both soil moisture and vegetation index over agriculture, whereas over the savanna-forest it responds more directly to vegetation index, as shown by the direction of the evaporative fraction color gradient. The 15 evaporative fraction is more variable over the agricultural field, explaining the less good fit of the regression ( $R^2 = 59$  %) compared to over the savanna-forest ( $R^2 = 66$  %). Inclusion of net radiation, cloud cover, down-welling radiation, and wind speed in this model did not significantly change the quality of the regression. This further supports our understanding that this is a moisture limitation of the system. This linear regression model provides and estimation that confirms our understanding of the physical basis of fluxes.

# **4** Discussion

# 4.1 Energy balance

The most striking observation is that the savanna-forest had consistently higher levels of both sensible and latent heat flux across all months (figures 4, 5). Sensible heat fluxes over the two surfaces showed the greatest difference in November and

- October (figure 6). Latent heat flux is the most different between the two land uses in August and May, transition times when the access to water in the catchment is not uniform (figure 6). The greatest similarity between the two land covers was during the wet season, May through September. The difference between the energy balance of the two sites was the most accentuated in the transition from the wet to dry season that occurred in the month of October for sensible heat flux and in the early wet season for latent heat flux (figures 4, 5, 7). Because of this observation, we can expect land use changes to have
- the most impact during these transition periods, due to differences in growth patterns and rooting depths. In particular since

agricultural crops are planted, their germination and development is determined by agricultural decisions above water and energy availability. The behavior during the growing season from June through August varied so dramatically between 2009 and 2010 because in 2009 short season millet was planted whereas in 2010, it was left fallow. We can imagine that this difference is due to the high growth rates of the agricultural crop through August and the subsequent harvest, where as the

- 5 fallow crops grow quickly, transpiring the most in July and then stabilizing into August (figure 4). The sensible and latent energy fluxes from the fallow were lower that that of the forest in 2010, whereas when short season millet was growing, they were more similar. By comparing the changes in soil moisture along the diurnal cycle, we see that even in the dry season there is some variation even in the dry season, particularly at shallow depths, thus in contrast to other authors we cannot attribute the early peak purely to stomata behavior during the dry season (Mamadou et al., 2016). In contrast, during the rainy season, it declined, following the cycle of available radiation.
- The net radiation was very similar over both land surfaces, with the greatest differences occurring in the dry season. However early in the dry season net radiation was higher over agricultural whereas early in the dry season, it was higher over forest-savanna. This second, counter intuitive finding, is supported by other work in the subregion that found that net radiation was higher over woody vegetation during the dry season and explained the difference because surface tempaerature
- was lower and net long wave radiation was higher (Guyot et al., 2009). The residual of the energy budget showed fewer clear patterns, but across seasons, it was greater over the field, likely due to larger ground heat flux into the bare soil. Two contrasting trends explain why the sensible heat fluxes becomes less similar as the year progresses whereas the latent heat fluxes becomes more similar. First, at the start of the year, the agricultural field is covered with bare ground and the rocks are exposed above the forested area, and throughout the growing season, the bare ground is progressively covered with
- grass whereas on the hill, the rocks remains exposed. At the end of the wet season, the grass senesces and remains until it is burned in late December or January. The contrast of the annual cycles of bare ground and bare rock drives the difference in sensible heat flux. The bare ground has a high level of albedo compared to the rocks, creating a difference in available energy for turbulent fluxes.

Second, at the start of the year, the upper trees have access to water coming from the springs at the base of the rocks. Although the level of water availability and vegetation increases during the wet season and declines during the following dry

season, the spring is permanent perhaps due to subsurface lateral water transfer that also produces shallow groundwater that is available to the trees (Mande, 2014). In contrast, the water availability in the field and the corresponding greenness closely follow the annual cycle of precipitation, driving the variation in latent heat. The ephemeral stream stops flowing at the end of December, when the grasses dry up and the latent heat flux returns to being drastically different between the two sites

(Mande, 2014)..

Our observations of H and  $L_eE$  are higher than fluxes previously measured in the region (Bagayoko et al., 2007; Dolman et al., 1997; Gash et al., 1997; Guichard et al., 2009; Guyot et al., 2009; Mauder et al., 2006; Schüttemeyer et al., 2006; Timouk et al., 2009), which can be explained by its location inside a semi-protected area with regionally relative high amount of vegetation, increasing  $L_eE$ , and an abundance of rocks, raising H. Furthermore, annual cycles of ratios between H and  $L_eE$ 

vary by wind sector (figure 6), demonstrating that small variations in land-cover, topography, and moisture availability can lead to dramatic differences in evaporation and evaporative fraction. This is consistent with local land management philosophy, which emphasizes the importance of maintaining the gallery forest, and springs therein, as a common moisture reservoir in the dry season and in case of drought. More continuous, long-term measurements during extreme years would

- reinforce the validity of this local belief. Our results also emphasize that the forest, even though it is primarily an open wooded savanna, has a higher level of productivity than the rain-fed, hand-farmed fields. Our values of sensible and latent heat flux are most similar to those measured in Ejura, Ghana in 2002 (Schüttemeyer et al., 2006). Ejura is about 500 km from our site, and though quite far and typically placed in a different category of climate, we measure similar values compared to other areas of West Africa (Guyot et al., 2009), perhaps because measurements took
- place in November, when vegetation there would be most similar to that at our site. Kompienga is the most similar and closest to ours, though overlapping in terms of seasons, but measurements are still lower than ours (Bagayoko et al., 2007). The scale incongruity between the turbulent flux (sensible and latent) flux measurements and the net radiation may explain our lack of closure, instead of, for example surface heat water storage during floods (Guyot et al., 2009). There is a strong topographic difference close to the forested area, however we are confident that our planar fit correction effectively corrected
- for the corresponding effect on turbulent fluxes.

The high magnitude of turbulent fluxes can be explained by Tambarga-Madjoari's location in the midst of a natural park and hunting concession as well as our measurement of a gallery forest, which would demonstrate the importance of the nearby vegetation cover. On the whole, our measurements are comparable to those elsewhere in West Africa, given that most of these sites (Mali, Niger) are more sahelian, and thus have less moisture availability, and others (Nigeria, Benin, Ghana) are considerably further south and thus have denser vegetation (White, 1986).

#### 4.2 Evaporative Fraction

The median monthly evaporative fraction in the forest savanna was lowest in April at 0.47, and highest in September at 0.77, and for the agricultural fields, it was also lowest in April at 0.31 and highest September at 0.79. These values are in general higher than previously found in similar environments during the rainy season (Bagayoko et al., 2007; Brümmer et al., 2008;

- Guyot et al., 2012; Mamadou et al., 2016). In contrast, our site has a lower dry season evaporative fraction than nearby sites, which suggests that moisture cycling during the wet season is more complete, and that there is less storage into the dry season. This is likely due to a combination of vegetation, soil, and topographic characteristics. We can compare our values of evaporative fraction with other environmental conditions. Over West Africa, the self-
- preservation concept of the evaporative fraction could be used together with variables such as albedo, temperature at the 30 surface, and a vegetation index to obtain a reasonable estimate of evaporation (Compaore, 2006). A high correlation between mid-day evaporation and the evaporative fraction exists in Kenyan grasslands (Farah et al., 2004). Evaporative fraction might be affected by cloud cover which alters incoming radiation (Brutsaert and Sugita, 1992), however cloudiness is not related to the stability of evaporative fraction in 2005-2006 in Brazil (Santos et al., 2010). Evaporative fraction may also be

related to other environmental parameters that are increasingly available and reliable which are obtained remotely through satellites, such as soil moisture (Crago, 1996; Hall et al., 1992). One limitation of this approach is that of the temporal scale since evaporative fraction is computed by day but surface conditions and moisture availability can change instantaneously. However it is extremely valuable to be able to estimate evaporative fraction when only NDVI, or other remotely sensed

5 parameters are available. For this reason our estimation of evaporative fraction with a simple model based on the physical basis of fluxes linked to vegetation and soil moisture, is useful (figure 9). This model can be refined with longer and more varied datasets to increase its accuracy and suitability regionally.

#### 4.3 Social Context

- The gallery forest over which we measure fluxes includes two springs that were the main water source for the village at its founding until a generation ago (Ceperley, 2014). Its important role in local history means that special institutions existed in local tradition for its conservation. For example, wetlands, and other areas with abundant water, fall into a gourmantche category for land called "Tinjali" or land that was not farmed because of cultural taboos until recent development projects and the introduction of rice farming (Swanson, 1978). In addition, since it is forested, it is protected because gourmantche believe that trees have spirits, which may prohibit them from being cut or used, and that some are considered good and some
- 15 bad. So it is reasonable to conclude that the presence of the forest is not only because the water availability provides the habitat, but also because the village has protected it on some level. Additionally, it is reasonable to think that the institutions that protect this forest are not only protecting it, but also the ecosystem services that it preserves. Our work suggests that these ecosystem services include the cycling of moisture into the atmosphere and the eventual generation of rainfall. In this light, gourmantche myth explains that there is a sack of water above the atmosphere that spirits can pierce, bringing rain
- 20 delivered by clouds (Alves, 2012). If the gods pierced those clouds with "trees" then our research seems to be right on target in terms of validating what traditions have long known. This is an important tool, as land use becomes more and more contested, the validation of local institutions or land uses can ensure the continuation of these practices.

#### **5** Conclusions

- Sensible and latent heat fluxes were higher over a savanna-forest than a semi-cultivated (millet-fallow) field according to our measurements in a sudanian ecosystem of West Africa in 2009 and 2010. The sensible heat and latent heat flux are generally higher over the savanna-forest because of its more permanent water availability and corresponding greenness, higher productivity and the amount of rocky terrain. For example, the diurnal cycle of latent heat flux peaked earlier in the day during the dry season in the agricultural field, suggesting depletion of all available moisture by late morning. These observations of sensible and latent heat flux are higher than fluxes previously measured in the region, potentially due to this site's location inside of a semi-protected area. Analysis of wind sectors separately revealed that particular sectors,
  - corresponding to the location of particular features, for example over the ephemeral wetland and the gallery forest,

contributed higher amounts to the flux. Local land management emphasizes the importance of maintaining the gallery forest, and springs therein, as a common humidity reservoir in the dry season and in case of drought. Changes in land cover may even have consequences for local rainfall triggering, causing cascading effects that transform both the energy and water budgets.

- 5 Continuous, long-term measurements during drought and moist years are essential to prove the long-term validity of our observations. Additionally, variations in exchanges according to small landscape features could result in enormous underestimation for up-scaling. We recommend using eddy-covariance measurements such as these to improve estimates with more easily maintained and obtained meteorological station and satellite data. The evaporative fraction is dependent on NDVI, which is an important finding for modeling and up-scaling. Efforts focused on preserving hydrologic services need to take anomalies into account and reinforce cultural institutions that protect wetlands and gallery forests.
- The development of a simple NDVI based indicator of evaporative fraction is transferable to other semi-arid systems, agroforestry parklands, and open wooded savannas around the globe. Globally, the African continent and often semi-arid environments represent a gap in observations of land atmosphere interactions. This study is an important step in filling this gap and proposing a tool with still greater potential.
- 15 Our results point to the necessity for ground measurements for eventual up-scaling from point to regional evaporation measurements in remote and less-studied regions of the globe. We began this work with discussion with community partners and to bring it full circle we conclude this paper by relating it back to the cultural context.

#### **6** Acknowledgements

The Velux and 3rd Millennium foundations funded a large part of this research in addition to support from the young researcher's (KFPE) grant from the Swiss Agency for Development and Cooperation. Alexandre Repetti, Jean-Claude Bolay, and the center for cooperation at EPFL initiated and continued to this project. We would like to thank all of our colleagues, assistants, and students based both in Switzerland in Burkina Faso who helped us with fieldwork. In particular, collaboration with the International Institute of Environment and Water Engineering facilitated this research. We give additional gratitude to the commune of Madjoari, its residents and its government, who hosted us and our equipment for the duration of this

25 project. This manuscript was greatly improved by feedback from reviewers and the scientific community, for that we are greatful. The final analysis was completed with the support of the NSERC discovery grant.

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
