# Peer review of "Evaporation from Cultivated and Semi-Wild Sudanian Savanna in West Africa"

_Hydrology and Earth System Sciences, 2016_

## Referee Comment (RC1) · Anonymous Referee #1 · 16 Jan 2017

Summary: This paper studied evaporation and more broadly the surface energy budgets at two sites of different land cover types (Savanna v.s. Agriculture field) in semi-arid West Africa. Observations spanning from May 2009 to September 2010 were used. Characteristics and the controlling variables of evaporation were identified, which is important for such an area lacking of experimental data. The comparison between results over two land cover types is also an interesting contribution to the literature. I also like the connection to the social and culture aspects. Overall the paper is well written and presented. I only have one major comment and some minor comments and thus recommend minor revision.

Major comments:

1, I find the calculation of net radiation for all the meteorological stations does not really

contribute to the paper. If I understand correctly, the measured net radiation is not really used except in validation against the measured ones at the energy balance stations. It is interesting as a stand-alone piece but is not well integrated into this paper. And I find removing this does not change the paper significantly or even make the paper more focused. One supporting evidence for my argument is that none of the related calculations was even mentioned in the abstract and conclusions.

Another reason I think this should be removed is that the calculations were not studied deeply. For example, I think the explanation of biases in the calculated net radiation is not thorough. The biases in the net longwave radiation in Figure 3 are so large that I don't think it is due to the wavelength of the instrument.

Give these two reasons, I would strong suggest remove this part.

Minor comments:

1, line 23, page 8: I would not use consistent. Maybe 'smooth'.

2, line 22-25, page 9: This should go to a place much earlier when the radiative and flux measurements are first discussed (for example in Figure 4).

3, Figure 7 does not reveal much new information and should be removed. On the other hand, the discussion of Figure 8 is rather superficial at this moment. I only see 1 sentence for Figure 8 (line 20, page 9).

4, Figure 10 (section 3.3.1): R2 should be provided in order to claim "NDVI and soil moisture have the strongest positive correlation with evaporative fraction".

5, Figure 11 has too much non-essential and redundant information. The only helpful information is the middle panel showing the validation of the model. The others can be safely removed. On the other hand, Figure 12 can be combined with the middle panels of Figure 11. By the way, figure 12 is not even mentioned.

6, line 9, page 11: should be figure 9, not figure 8.

---

## Author Comment (AC1) · 6 Feb 2017

Dear Anonymous Referee 1, Thank you very much for your comments regarding our manuscript, "Evaporation from Savanna and Agriculture in Semi-Arid West Africa received and published on January 16, 2017.

1, I find the calculation of net radiation for all the meteorological stations does not really contribute to the paper. If I understand correctly, the measured net radiation is not really used except in validation against the measured ones at the energy balance stations. It is interesting as a stand-alone piece but is not well integrated into this paper. And I find removing this does not change the paper significantly or even make the paper more focused. One supporting evidence for my argument is that none of the related cal-

culations was even mentioned in the abstract and conclusions. Another reason I think this should be removed is that the calculations were not studied deeply. For example, I think the explanation of biases in the calculated net radiation is not thorough. The biases in the net longwave radiation in Figure 3 are so large that I don't think it is due to the wavelength of the instrument. Give these two reasons, I would strong suggest remove this part.

This is understandable; it arose from a discussion among co-authors. I will move this figure and the discussion surrounding it to the supplementary material.

Minor comments: 1, line 23, page 8: I would not use consistent. Maybe 'smooth'.

Thank you for this suggestion, this has been changed.

2, line 22-25, page 9: This should go to a place much earlier when the radiative and flux measurements are first discussed (for example in Figure 4).

This is a good point; it has been moved to the discussion of figure 5 on the previous page.

3, Figure 7 does not reveal much new information and should be removed. On the other hand, the discussion of Figure 8 is rather superficial at this moment. I only see 1 sentence for Figure 8 (line 20, page 9).

More discussion has been added for figure 8. Figure 7 and related discussion has been added to supplementary material.

4, Figure 10 (section 3.3.1): R2 should be provided in order to claim "NDVI and soil moisture have the strongest positive correlation with evaporative fraction".

A table of R2 values has been added to the figure.

5, Figure 11 has too much non-essential and redundant information. The only helpful information is the middle panel showing the validation of the model. The others can be safely removed. On the other hand, Figure 12 can be combined with the middle panels of Figure 11. By the way, figure 12 is not even mentioned.

The validation of the model and figure 12 have been combined and discussion has been developed around figure 12.

6, line 9, page 11: should be figure 9, not figure 8.

Thank you, this has been changed.

Sincerely,

Dr. Natalie Ceperley, University of Lausanne

Please also note the supplement to this comment:
http://www.hydrol-earth-syst-sci-discuss.net/hess-2016-672/hess-2016-672-AC1-supplement.zip

a)

b)

c)

d)

e)

| | Evaporative Fraction | |
|---|---|---|
| | Agriculture | Savanna-Forest |
| Vegetation (NDVI) | 69.76 | 74.71 |
| Soil Water content | 64.63 | 48.33 |
| Net Radiation | -6.88 | -13.33 |
| Wind Speed | -51.85 | -43.07 |
| Cloud Cover | 19.17 | 38.94 |

- Agriculture
- Savanna-Forest

**Fig. 1.** Daily evaporative fraction over study period compared with relevant variables. The least squared regression lines are shown for each plot. Correlations between variables are in the table.

Agricultural

$ef_{ag}=1+0.41*ndvi+1.4*vwc$ $(R^2=0.59)$

Forest

$ef_{gf}=1+0.34*ndvi+0.48*vwc$ $(R^2=0.66)$

Agriculture

Savanna-Forest

—— VWC+NDVI    —— NDVI    · Measurements

**Fig. 2.** Further Examination of relationship between Soil Moisture, Vegetation Index, and Evaporative Fraction. Upper plots show quality of fit and lower plots show modeled evaporative fraction.

---

## Referee Comment (RC2) · Anonymous Referee #2 · 9 Feb 2017

Thank you for giving me the opportunity to review the paper "evaporation from Savanna and Agriculture in Semi-Arid West-Africa". The paper presents 17 months of energy fluxes data over the two dominant land covers found in the Sudano-Sahelian area of West-Africa, namely fallow/crop plots and woody savannah. Such data are highly valuable because they are very limited in the area, yet in this specific region the surface to atmosphere exchanges are complex to characterize while fundamentals for the understanding of both atmospheric processes and hydrological cycle. In the context of exceptionally high demographic rates and subsequent fast-occurring land use changes, there is a strong need to both document and monitor over long time surface fluxes for different land covers. This study is fully dedicated to that scope. The authors also seek to derive simpler estimates of evapotranspiration from evaporative fraction (EF) as a first step to obtain spatial evapotranspiration from remote-sensing products.

Overall, I find the dataset presented of significant importance and the paper well written, and I do recommend publication in the journal. I do, however, think that there is currently too much ambition in the paper which somehow undermines its relevance. I see two main points currently developed in the paper: 1) the physical interpretation of fluxes data backed up by additional meteorological/hydrological/environment variables and 2) the development of a proxy-based method for regionalizing evapotranspiration based on EF. While the two aspects are given equal importance in the abstract, the first one takes a significant proportion of the paper, but could still deserve more development and discussion (see comments below). Some of the major shortcoming of this aspect is –to my point of view- the limited discussion with respect to other similar, studies undertaken in the area, and a slightly more comprehensive footprint analysis to discuss the high fluxes observed. On the other hand the second aspect is only limited to deriving an empirical relationship between EF and NDVI and soil moisture, and without providing an evapotranspiration map, for instance, which could be a nice achievement. The easiest way to deal with this point is probably to modify the abstract and develop some aspects of the physical basis of E fluxes, but I encourage the authors to develop more thoroughly the NDVI-based model (although so far, the EF derived time series seems unrealistically high to me).

Due to the large amount of minor comments that I have, I do recommend a major revision, but I think that the authors will have no trouble tackling all the little issues regarding to their dataset. I am not a specialist on energy budget analyses, and my comments on physical interpretation will therefore be limited, although I have the feeling that they could be strengthened.

I do appreciate the link with social aspects, although much care should be taken when interpreting local practices without dedicated studies. I am not an English native speaker, so I won't be commenting much on the English, and please excuse my poor English. And last but not least, this is a nice study, thank you.

Below are my comments:

Title: I see two minor issues with the title:

o Savanna is a land cover –and as such a source for ET-, but agriculture is a practice, and therefore not similarly related to ET. I would recommend using "Savanna and a cultivated area" or similar. I am not a native English speaker, and this might not be so important, hence feel free to take this comment into account or not.

o The area is really on the edge of what can be considered as semi-arid. I would rather relate it to its eco-hydro-climatic classification of the transition zone between the sahelian zone and the Sudanian zone: the sudano-sahelian area. (Sahel is roughly comprised between 100 and 700mm annual rainfall, Sudanian area between 700-1400mm, and Guinean >1400mm; and some authors consider a sahelo-sudanian or sudano-sahelian transition band, which I think this region could fall into. Sudanian works too.

Abstract:

o L 20: I am not sure that you actually showed evidences that the fact that fluxes above the savanna-forest where higher was due to the number of rocks and trees and tree productivity. A comprehensive footprint analysis would, at least, be necessary.

o L25: I think a paragraph is missing in the paper, as Figure 12 is not explained, and it is never written in the main text that NDVI only is sufficient to predict EF. This should be also further discussed.

p2.L2: This is not straightforward: it should be sustained by a proper reference or moderated by adding "for example". For instance, there could be among-species differences in leaf renewal timing for some trees, not necessarily related to moisture availability.

P2.L18-25: I would move these equations to section 2.3 (as is done with EF- or move EF equation to the introduction, although this relates more to "measurements and calculations" than to an introduction).

P2.L26-32: consider removing some references. 2-3 references by statement are

enough.

P2.L32: Let me suggest a few references that should be carefully studied and probably discussed at the end of the article, because they are located either in similar eco-climatic context, or bounding (either in the south or in the north) the study area. (Brümmer et al., 2008; Guyot et al., 2009, 2012, Lohou et al., 2010, 2014, Mamadou et al., 2014, 2016; Velluet et al., 2014)

P3.L6: It is not the evaporative fraction which is based on the concept of self-preservation, but "the method that estimates daily evapotranspiration from evaporative fraction"

P3.L15: This has been studied in West-Africa and for a broad range of eco-climatic contexts by (Lohou et al., 2014)

P3.L19: is "raised" plateau an appropriate term? I have never heard it.

P3.L20: This is not true. I have not checked all the literature, but at least (Guyot et al., 2012; Lohou et al., 2014; Mamadou et al., 2016; Velluet et al., 2014) have studied energy fluxes in the area for different land covers and several seasonal cycles.

P3. L24-L27: I would move that to the conclusion.

P4. L 16-17: I am not sure this is relevant here.

P4. L18: It is not the sudanian area which is defined by alternating wet and dry season, but rather the monsoon climate. My suggestion would be (if you choose to keep sudanian instead of sahelo-sudanian): "The watershed falls within the sudanian zone of the west-African monsoon climate system, with alternating dry and wet seasons with the rainfall falling. . .."

P5.L7-8: This is an important remark: why not using ground T° being recorded with the Decagon moisture probes for calculating G? Only two depths are enough for a first calculation using the Harmonic method (Guyot et al., 2009). This would allow to be

less elusive about the residuals of the energy budget, and to bring more evidence to conclude on higher/lower G under fields/Savanna.

P6.L6: I really think that individual times series should be shown, because according to Figure 5, they can be much different. At least one for the forest-Savanna cover and another for the field cover, and for the two depths, on a same figure containing H, LE and RN for the two land covers and for the same period. Maybe NDVI too?

P6-7: section 2.8; I do agree with Reviewer #1 who proposes to remove this section. It could bring some interesting results, but would need much more developments, which are not the purpose of that paper. This would allow to gain some room for further developments in the text.

P8.L10. To me, the NDVI for the forest in the dry season is not as much greener as you would expect from a forest dominated by Vitellaria paradoxa, which only need about a month for leaf renewal. The NDVI pixel must include some significant herbaceous cover. If possible, could you estimate the vegetation classes distribution within the pixel?

P8.L17. These are not average diurnal cycles, but single days samples. For me, all the results and conclusions drawn from this analyze (otherwise very interesting!) are weak due to the undersampling of diurnal cycles. For instance, we could not state from these single days, that "Net radiation is slightly higher during the wet season (P8.L21)", (although this is probably true). If there is enough data to produce composite diurnal cycles over larger periods, please do. It is very hard to understand why there is such a significant LE flux in april over the field, which should be composed of bare soil according to the text and pictures. I can only suspect that there has been a rain in the previous day(s), - and there has been rain events in april according to Fig.4, and that the single day sample is too particular to be analyzed as a representative day. There may also be some other processes acting, such as lateral subsurface water transfer, and that should be discussed, but I doubt so.

Another solution is to provide time series of RN, LE and H (and SM, rainfall, NDVI), as proposed on the comment P6.L6, on a larger plot than Figure 8, and not separated by wind sectors.

These results should be discussed in the light of previous results for diurnal cycles obtained in the area (Guyot et al., 2012; Mamadou et al., 2014, 2016), for instance.

P8.L25: there is also, for both sites, this interesting feature in the late afternoon that LE>RN and even slightly positive during the night, as noted in (Mamadou et al., 2014, 2016), for instance.

P8.L31: It was lower over the field (if I am not mistaken).

P8.31: again, the expression "for all months" does not hold when only two days are compared.

P9.L2: this is an interesting statement, but it needs to be further supported. For instance, (Mamadou et al., 2016), for a forested site in similar conditions, noted an early LE peak in the dry season, but also that Soil moisture changed very little from the morning to the afternoon (<0.1mm). They concluded that it was probably a temperature limitation of stomatal opening.

P9.L2-7: this is not supported by a figure. Again, I think that time series of the fluxes should be shown. Also, I think the discussion on limitations should consider here conductances, which can be affected by several processes, such as moisture limitations, but also shading, or stomatal resistances.

P9.L5. What is the "absolute maximum"?

P9.L11. I am not sure about this conclusion. I think that dynamic aspects should be taken into account, as fields and savanna will not necessarily have equal amounts of RN at the same time. I am not entirely sure on this, and I think that calculating G from temperature probes could help on that.

P9.L15-19. I think this is similar to what is found in (Guyot et al., 2009)

P9L20-28: This relates to the footprint concept. To my point of view, the footprint should be explicitly calculated for specific periods. But I would understand that it could significantly impact the paper in its current form. At least, wind sectors of Figure 8 should be more discussed; For instance, I do not clearly understand what is the difference between all the time series of, say, the Field panel, how have they been calculated? Also, I may be wrong, but I guess that N-W winds are the Harmattan winds, bringing hot, dry, and dusty air from the Sahara, and S-E winds would be the monsoon inflows, bringing moist air? If so, and as this has significant impact in the resulting energy budget, it could deserve some further insights (see e.g. (Guyot et al., 2012)).

P9.L25-27. I agree with Reviewer #1 that Figure 7 does not bring much, thank you for agreeing to move this figure to the appendix. Also, I am not sure that such correlation is an evidence for a causal effect.

P10.L2. Burning practice should be defined in the site description.

P10.L11-15: thank you for the correlation coefficient added on this figure.

P10.L25: My guess is that there are serious temporal limitations in this approach, and although the correlation coef are not too bad, the approach could probably be improved by considering different time windows. For instance, in the field, there is bare soil in the dry season, and for a little while, only soil evaporation may occur, and there is probably no need to take NDVI into account. On the opposite, once the herbaceous layer has grown, the system may not be water limited anymore, and soil moisture is not needed anymore. During the growing phase only, the equation could probably produce better results.

Also, what soil moisture time series have you used? If this is the catchment averaged one, then the higher regression slope on soil moisture for the field further indicate a

higher moisture-controlled system. If the ultimate goal is to compare it with remote sensing-derived soil moisture, this could easily be added here.

P11.L4-10: some statements need to be made clearer (but maybe this is due to my poor english):

o 'sensible heat flux showed the greatest diff. in November & March': agree for November, although October seems even more different, but the ratio in March is close to 1. Unless the two mentioned surfaces are the two field surfaces (2009: millet, 2010 fallow)? But then it should be June and September.

o Again, for LE flux: the strong difference between forest and field are in March. I guess you point to the millet and fallow land use? Could you further discuss on why the fallow evapotranspirate less?

o L7-9: I think again Figure 8 is not clear enough, why not having a plot ( as suggested above) with the fluxes of forest + field on a similar panel? There could be a panel of Rn, of LE, H, Soil moisture(+precip), and, say, NDVI?

o L9: The strongest difference could also be expected in the dry season, no? (see e.g. (Mamadou et al., 2016)) Or please expand on why this is expected.

o On Rn: (Guyot et al., 2009) described a higher Rn on the dry season over woody cover, because surface temperature was lower, and LWnet higher. This could be discussed (if judged relevant), because there is an opposite behavior here. Although according to Fig.7 LWnet is higher in the forest in the dry season, and there are two tails in the LW-net box plot of Figure 6.

P11.L15-16: I am not sure why rock presence should be mentioned here. Plus, they are not the only producers of H fluxes. Maybe an extra sentence could be added? And to clearly show that they are somehow responsible for a good part of H flux, a footprint analysis should be undertaken.

Also, according to Figure 1, there is a strong topographic difference close to the

forested area. I am not a specialist, but could that affect fluxes in any way?

P11.L18: this is not clear to me: are we talking about trees getting water from open waters in the channel? If not, it should be simply stated that upper trees have 'probably' access to shallow groundwater at the origin of the springs.

P11.L20: what is permanent?

P11.L20-21: I guess reference to the land cover is missing: this sentence relates to the field, right?

P11.L26. According to the picture in Figure 5, the field location is not particularly covered with more vegetation. The LE fluxes there are really high, and I have no explanation for this. On a Large Aperture scintillometer beam pathway, including very mixed cover, and including also a gallery forest preserved for similar reasons (Guyot et al., 2012) had much lower LE fluxes for instance. Same observation for (Brümmer et al., 2008) in South West Burkina Faso, at a similar latitude, or (Mamadou et al., 2016) in a fully forested parkland in North Benin. This is very surprising, and I have no explanation.

I am also curious, I have looked at the satellites images in google map, and the field location seems to appear in a middle of a much larger forest. What is at the origin of this specific deforestation? (this is not relevant in the review, only personal interest).

P12.L5-8, this should be rewritten according to the references given in the comment P2.L32. There is a site in Burkina Faso in similar climatic context, and sites in Benin are not that far south, and as for the cultivated ones, have probably a rather similar ecological context.

P12.L11-18: I have read that references to figure 12 have been added, as well as more discussion. Currently there is a strong issue that really needs to be addressed: The EF clearly shows that fluxes are very high, and this should be commented. EF could be compared with the study of (Lohou et al., 2014), which spans different eco-hydrological
contexts through the N-S West-African gradient. In your Figure 12, EF does not seem to drop below 0.5, which is exceptionally high. If I roughly take a yearly average value of 0.6, and a yearly average daily value for ETo of say 5mm (Wang et al., 2007), which is probably a lower bound for ETo in this area, this produces about 1100mm or evapotranspiration yearly, which is much higher than precipitation. Footprint analysis could be undertaken to identify the reason for such high evapotranspiration. This could lead to very promising result.

P12.L23: cultural taboos, or health-related issues (malaria)?

P13.L8: Add "in the field", to make it clearer

Figures:

In general, figure captions are too long and should be limited to description and not interpretation, but Figures are of good quality.

Figure 1: If possible, I would prefer the elevation shown as contour lines to better see the satellite image below. For instance, it is impossible to locate the village. And also, it is not easy to understand what is the dark line S-E / N-W just north of the study area: is it a plateau edge or a riparian area? I would recommend to put contour lines of the whole map, and draw the catchment contours with a proper shape.

Remove "land locked country" and 100m of elevation change. . .and the plateau". If this is important, it should be put in the main text.

Figure 2: Figure caption: you could remove the last sentence, which is interpretation

Figure 4: soil moisture is exceptionally high in 2011 (known to be a dry year in the area, so I doubt there is any issue with the rain record), could you comment on that?

Figure 5: thank you for putting in the same figure both photographs and diurnal cycles, this makes a nice figure.

Figure 6: Figure caption: you could remove interpretation sentences. I also think there

[Figure]

is some confusion in the middle-bottom two description.

Below are some technical corrections: P2.L1: change sudanian to sahelo-sudanian P2.L16: add "in" before Eq.(1) P3.L7: [...]the diurnal cycle "of" each [...] P4. L3: [...] were made in "a" small [...] P4. L21 "in the" Guinean zone P6.L9. remove "infrared" P7.L4. where "smoothed" P8.L5. remove first occurrence of "air". P8.L21. for the both sites? P8.L25. Heat instead of heath P9.L12: remove "below" P9.L21: contributed "to" higher fluxes? P10.L24: be fitted? P11.L16: at the end of the "wet" season? P12.L27: has protected "it"?

Thank you again, this is a nice data set, and nice work.

References:

Brümmer, C., Falk, U., Papen, H., Szarzynski, J., Wassmann, R. and Brüggemann, N.: Diurnal, seasonal, and interannual variation in carbon dioxide and energy exchange in shrub savanna in Burkina Faso (West Africa), J. Geophys. Res. Biogeosciences, 113(G2), G02030, doi:10.1029/2007JG000583, 2008.

Guyot, A., Cohard, J.-M., Anquetin, S., Galle, S. and Lloyd, C. R.: Combined analysis of energy and water balances to estimate latent heat flux of a sudanian small catchment, J. Hydrol., 375(1–2), 227–240, doi:10.1016/j.jhydrol.2008.12.027, 2009.

Guyot, A., Cohard, J.-M., Anquetin, S. and Galle, S.: Long-term observations of turbulent fluxes over heterogeneous vegetation using scintillometry and additional observations: A contribution to AMMA under Sudano-Sahelian climate, Agric. For. Meteorol., 154–155, 84–98, doi:10.1016/j.agrformet.2011.10.008, 2012.

Lohou, F., Saïd, F., Lothon, M., Durand, P. and Serça, D.: Impact of Boundary-Layer Processes on Near-Surface Turbulence Within the West African Monsoon, Bound.-Layer Meteorol., 136(1), 1–23, doi:10.1007/s10546-010-9493-0, 2010.

Lohou, F., Kergoat, L., Guichard, F., Boone, A., Cappelaere, B., Cohard, J.-M., Demarty, J., Galle, S., Grippa, M., Peugeot, C., Ramier, D., Taylor, C. M. and Timouk, F.:

Surface response to rain events throughout the West African monsoon, Atmos Chem Phys, 14(8), 3883–3898, doi:10.5194/acp-14-3883-2014, 2014.

Mamadou, O., Cohard, J. M., Galle, S., Awanou, C. N., Diedhiou, A., Kounouhewa, B. and Peugeot, C.: Energy fluxes and surface characteristics over a cultivated area in Benin: daily and seasonal dynamics, Hydrol Earth Syst Sci, 18(3), 893–914, doi:10.5194/hess-18-893-2014, 2014.

Mamadou, O., Galle, S., Cohard, J.-M., Peugeot, C., Kounouhewa, B., Biron, R., Hector, B. and Zannou, A. B.: Dynamics of water vapor and energy exchanges above two contrasting Sudanian climate ecosystems in Northern Benin (West Africa), J. Geophys. Res. Atmospheres, 2016JD024749, doi:10.1002/2016JD024749, 2016.

Velluet, C., Demarty, J., Cappelaere, B., Braud, I., Issoufou, H. B.-A., Boulain, N., Ramier, D., Mainassara, I., Charvet, G., Boucher, M., Chazarin, J.-P., Oï, M., Yahou, H., Maidaji, B., Arpin-Pont, F., Benarrosh, N., Mahamane, A., Nazoumou, Y., Favreau, G. and Seghieri, J.: Building a field- and model-based climatology of local water and energy cycles in the cultivated Sahel – annual budgets and seasonality, Hydrol Earth Syst Sci Discuss, 11(5), 4753–4808, doi:10.5194/hessd-11-4753-2014, 2014.

Wang, Y.-M., Traore, S. and Kerh, T.: Determination of a Reference Model for Estimating Evapotranspiration in Burkina Faso, in Proceedings of the 6th Conference on 6th WSEAS Int. Conf. On Artificial Intelligence, Knowledge Engineering and Data Bases - Volume 6, pp. 211–216, World Scientific and Engineering Academy and Society (WSEAS), Stevens Point, Wisconsin, USA. [online] Available from: http://dl.acm.org/citation.cfm?id=1348485.1348523 (Accessed 9 February 2017), 2007.

---

## Short Comment (SC1) · 13 Mar 2017

The paper by Ceperley et al. describes a year and half of energy fluxes in two eddy covariance sites (forest and agricultural field) of a semi-arid, mixed-used catchment in West Africa. The material is appropriate for a scientific study and the data obtained "appear" to be high-quality. The work seems interesting and worthy of publication in HESS journal however, before it can be accepted for publication a major revision is required.

Major Comment: Firstly, the paper is poorly organized, with much extensive content, though not exhaustive and too much interpretation in the results. While the methodology, in particular the eddy covariance data treatment requires a particular attention to have reliable turbulent fluxes, this was partially presented by the authors and the units

of keys variables were omitted. In addition, there was a total confusion in the signification of such variables. For example, the available energy is not the sum of turbulent fluxes (H+LE) but rather the difference between the net radiation and the soil heat flux (Rn-G), see L13, L25, p5; and section 3.2.2.

Secondly, one of the main points of this paper was in the site comparison; however, basic information about the research sites was lacking. Did both sites have similar soil characteristics? The large differences in soil water content may indicate site differences in soil texture. Also, more information is needed about the flux footprint. What was the fetch? Was the vegetation in particular (the rain fed site) within the flux footprint homogenous? The forest site seems to be located in a very complex topography according the map of the site (Figure 6). How this has been taken into account in the analysis of eddy covariance data? These aspects are important for understanding and interpreting the results. Finally, what is the value of the slope? Some of the writings throughout the text may be rewritten in more compact and yet concise style without losing the message they want to convey to the readers. Some conclusions are drawn without the support of data. Details can be found at the specific comments.

Specific comments

Abstract L18 – 20: Which period of the year? I am very surprised with this result! L18 – 22: I don't agree with this deduction. The presence of rocks and trees cannot, from my point of view, allow you to say that the soil heat flux is higher in the fields. You should take care with this assertion since you don't have any in situ measurements or direct calculation/estimation of this term of the energy balance to reinforce your conclusion.

Introduction L15 – 25: Give the units of all variables the first time that they are used. . L14 – 16: The cited reference Foken, 2008 "The energy balance closure problem: An overview" is not an appropriate reference. T. Foken has never worked on the link between global atmospheric processes and the land surface atmosphere interaction. Please provide an appropriate reference. L18: Write LeE is "latent heat flux" instead

of latent energy flux. L29: Replace Evaporation by Evapotranspiration since you were talking about vegetated surface. L30 – 33: It is true that in situ data of energy and water vapor fluxes were limited "in the past", but today there are a lot of studies which have been conducted in this part of Africa (Guyot et al, 2009; 2012; Mamadou et al., 2014 and 2016; Velluet et al., 2014; Timouk et al., 2009; Ramier et al., 2009).

L12 – 15, p3: THIS is not general but depends on the region; authors should specify the region in which this result has been obtained. L1 27, section 2.1 there are too much information's which are from my point of view not really essential for the interpretation of fluxes. An example "the village is made up of a majority...". I cannot get the importance of this sentence and elsewhere in the section 2.1. L32 p4: Infrared gas analyzers. Open path or closed path? Need to be precized L1 p5: Replace eddy correlation by eddy covariance. What is the distance between the two studied sites?

In Section 2.3 : L10: This is not true!! Sensible and latent heat fluxes cannot be measured at a half hour time step if you really used eddy covariance system to measure the fluctuation. How the sampled data have been then processed? What are the selection criteria? Given the complexity of these measurements, it is very shocking to see that certain details were not presented. Why do you use day light measurements (8am – 4pm) for the comparison? Give the reason of this choice. L20: Give the unit of different variables of Eq 4. L25: Here instead of using the day light measurements as mentioned in L10, you preferred to use midday average. Why? L9, section 2.6: What do you called the incoming shortwave infrared radiation? Is it the incoming shortwave radiation? If yes, make it clearer. P7, Give the units of variables of Eq 9 to Eq12. L5, p7: Longwave incoming or reflected radiation? L8: Write Ts instead of T s

Section 3.1, L5: I cannot get the meaning of this sentence. Section 3.2.1, L17: You started by saying that energy balance varied according to the month, i agree and now you compare a single day in April and in July. This is not coherent with the title of the section. L23: Write the sensible heat flux L24-25: In the sentence, "by July the latent heat has surpassed the sensible heath..". How do you explain this fact? Replace heath

by heat in the sentence. L26-27: What was the magnitude of this residual in the morning? residual is lower over the savanna (which values?) and what about the agricultural field? L30: What is the dust season? How was it objectively defined? Although the dust is something common over the region, it should affect also the measurement of the second site. How do you explain the fact that the net radiation was lower over the savanna-forest? L31-32: I am very surprised with these results. They are contrary to those obtained over the region...I would like to see the temporal evolution of H and LE based on half hourly data over the two studied sites. L1 – 4, p9: I cannot get the meaning of the sentence "The timing of the peaks of latent energy...the peak in the diurnal was after noon". This sentence may be rewritten in more compact and yet concise way. Replace latent energy by latent heat flux and elsewhere in the paper. L6- 7: I would like to see the diurnal cycle of the available soil moisture Section 3.2.2: What is the general correlation? L25: It is normal since the reflected longwave radiation depends on surface temperature (Eq. 9) Section 3.2.3: it is very surprising that the savanna-forest contributed more sensible heat flux throughout the year than the agricultural land. The convection above the agricultural field should be more than that of the forest because the "exposed area" and also the presence of vegetation over the forest which should limit this process.

Section 3.3.1 Could you give the values of coefficient correlation and their associated p-values? It seems that wind speed is also correlated with EF. How have you identified the two dominant variables? How landscape moisture availability can be expressed as both NDVI and soil moisture? Higher levels of soil moisture? Which levels? What is the total net radiation? In the sentence "Total net radiation does not show a strong influence, suggesting that this is not a radiation limited system", I do not see the data which support this conclusion. In L21, replace supposition by hypothesis. Section 3.3.2: Give the values of coefficient correlation and their associated p-values of your fit. Section 4.1, L13-16: I am not sure for these explanations... L19-20: What allow you to say that the level of water availability is permanent? Show then the water table in dry and wet season? L21: Replace latent heat by latent heat flux (and elsewhere in

the paper) L7-9, p12: I cannot get the meaning of the sentence In table 1: The Li-7500 measures both H2O and CO2 concentrations not "HO concentration". It seems that authors only provided the height of sensors above the agricultural field. What is the height of eddy covariance and additional measurements above the savanna-forest? Figure 4: Environmental parameters at study site (which one?) Figure 5: Write in the title Diurnal cycle of the energy balance components. In the title of Figure 6 : H+LeE is not the total available energy!!! Figures 11 and 12 are not cited in the paper.

References Ramier, D., N. Boulain, B. Cappelaere, F. Timouk, M. Rabanit, C. R Lloyd, S. Boubkraoui, F. Métayer, L. Descroix, and V. Wawrzyniak. 'Towards an Understanding of Coupled Physical and Biological Processes in the Cultivated Sahel-1. Energy and Water'. Journal of Hydrology 375, no. 1–2 (2009): 204–216. Timouk, F., L. Kergoat, E. Mougin, C. R. Lloyd, E. Ceschia, J. M Cohard, P. Rosnay, P. Hiernaux, V. Demarez, and C. M. Taylor. 'Response of Surface Energy Balance to Water Regime and Vegetation Development in a Sahelian Landscape'. Journal of Hydrology 375, no. 1–2 (2009): 178–189. Guichard, F., L. Kergoat, E. Mougin, F. Timouk, F. Baup, P. Hiernaux, and F. Lavenu. 'Surface Thermodynamics and Radiative Budget in the Sahelian Gourma: Seasonal and Diurnal Cycles'. Journal of Hydrology 375, no. 1–2 (2009): 161–177. Velluet, C., J. Demarty, B. Cappelaere, I. Braud, H. B.-A. Issoufou, N. Boulain, D. Ramier, et al. 'Building a Field- and Model-Based Climatology of Local Water and Energy Cycles in the Cultivated Sahel – Annual Budgets and Seasonality'. Hydrology and Earth System Sciences 18, no. 12 (10 December 2014): 5001–24. doi:10.5194/hess-18-5001-2014. Mamadou, Ossénatou, Sylvie Galle, Jean-Martial Cohard, Christophe Peugeot, Basile Kounouhewa, Romain Biron, Basile Hector, and Arnaud Bruno Zannou. 'Dynamics of Water Vapor and Energy Exchanges above Two Contrasting Sudanian Climate Ecosystems in Northern Benin (West Africa): Water Vapor and Energy Exchanges'. Journal of Geophysical Research: Atmospheres, 2016. doi:10.1002/2016JD024749. Mamadou, O., J. M. Cohard, S. Galle, C. N. Awanou, A. Diedhiou, B. Kounouhewa, and C. Peugeot. 'Energy Fluxes and Surface Characteristics over a Cultivated Area in Benin: Daily and Seasonal Dynamics'. Hydrology and

Earth System Sciences 18, no. 3 (7 March 2014): 893–914. doi:10.5194/hess-18-893-2014. Guyot, Adrien, Jean-Martial Cohard, Sandrine Anquetin, and Sylvie Galle. 'Long-Term Observations of Turbulent Fluxes over Heterogeneous Vegetation Using Scintillometry and Additional Observations: A Contribution to AMMA under Sudano-Sahelian Climate'. Agricultural and Forest Meteorology 154–155 (March 2012): 84–98. doi:10.1016/j.agrformet.2011.10.008. Guyot, Adrien, Jean-Martial Cohard, Sandrine Anquetin, Sylvie Galle, and Colin R. Lloyd. 'Combined Analysis of Energy and Water Balances to Estimate Latent Heat Flux of a Sudanian Small Catchment'. Journal of Hydrology 375, no. 1–2 (August 2009): 227–40. doi:10.1016/j.jhydrol.2008.12.027.

---

## Author Comment (AC3) · 7 Apr 2017

Dear O. Mamadou,

Thank you for your comments. I am sorry that you did not find my paper well organized and ready for publication. I will try to benefit from your comments after your close reading of my manuscript to improve it. You are right, that most of this work and my detailed literature review is a few years old, so as AR2 already suggested, I am in the midst of updating the comparison with other studies from the region, including your own. I am glad that you and AR2 could help me with this weakness.

Major Comment: Firstly, the paper is poorly organized, with much extensive content, though not exhaustive and too much interpretation in the results.

While the methodology, in particular the eddy covariance data treatment requires a particular attention to have reliable turbulent fluxes, this was partially presented by the authors and the units of keys variables were omitted. In addition, there was a total confusion in the signification of such variables. For example, the available energy is not the sum of turbulent fluxes (H+LE) but rather the difference between the net radiation and the soil heat flux (Rn-G), see L13, L25, p5; and section 3.2.2.

I will try to make my wording and variable reference more precise and detailed.

Secondly, one of the main points of this paper was in the site comparison; however, basic information about the research sites was lacking. Did both sites have similar soil characteristics? The large differences in soil water content may indicate site differences in soil texture.

I will include more information about soil characteristics.

Also, more information is needed about the flux footprint. What was the fetch? Was the vegetation in particular (the rain fed site) within the flux footprint homogenous? The forest site seems to be located in a very complex topography according the map of the site (Figure 6). How this has been taken into account in the analysis of eddy covariance data?

This was also a comment from AR2. We used a planar tilt correction to correct for the positive average wind speed as described by Oldroyd, H. J., Pardyjak, E. R., Huwald, H. and Parlange, M. B.: Adapting Tilt Corrections and the Governing Flow Equations for Steep, Fully Three-Dimensional, Mountainous Terrain, Bound.-Layer Meteorol., 1–27, 2015.

While I think a flux footprint would be intersting, I am saving it for a future analysis due to certain contraints that I discuss in my reply to AR2.

> These aspects are important for understanding and interpreting the results. Finally, what is the value of the slope?

I'll add a calculation of slope for the different wind directions around the EC stations.

> Some of the writings throughout the text may be rewritten in more compact and yet concise style without losing the message they want to convey to the readers.

This is a good comment, I will try to improve the writing.

> Some conclusions are drawn without the support of data. Details can be found at the specific comments.

I will try to take into account as many of your suggestions as possible. Your specific comments are very helpful and many of them will greatly improve my work. Thank you for taking the time to read everything so closely and I hope we have the opportunity to compare our work in more detail in the future.

Dr. Natalie Ceperley
* * *

---

## Author Response (AR1)

**UNIL** | Université de Lausanne
Institut des dynamiques de la surface terrestre
Catchment Hydrology
bâtiment Géopolis bureau 3317
CH-1015 Lausanne

Reed Maxwell
Editor
Hydrology and Earth System Sciences

Lausanne, 03.07.17

**Subject : Response to Referee Comments**

Dear Reed Maxwell,

Thank you for taking the time to coordinate the review and editing of our work. We apologize for our lengthy reply period and hope it hasn't disrupted your time schedule too much.

I have made the necessary changes to the article according to each of the reviewers and below I have included a point-by-point response to them. This point-by-point response does repeat many of my responses during the review phase. I will enclose both my marked up manuscript and a clean version.

In addition, according to my responses, I make some promises for future publications. These include calculating ground heat flux with the soil temperature measurements from decagon probes, a foot print analysis, an evaporation map, and vegetation class estimations for all pixels. Although this will undoubtedly fall beyond the scope of this review process, I will try to remember to inform you when my "distributed energy balance" calculation paper has been accepted for publication.

As Hans Peter Schmid (2002) describes, foot print analysis over vegetation and over uneven topography is difficult and hard to validate. This could be addressed in comparison with the calculation of evaporation from our distributed meteorological stations that we hope to publish soon. Similarly, an evaporation map of the catchment will fit well in this calculation as well. We could use these two results from this work and measurements in combination with our spatial measurements to determine how evaporation varies spatial and the footprint of our stations. This future publication would be dedicated to the spatial variation in evaporation.

Thank you again.

Natalie Ceperley on behalf of all coauthors

Point by point Replies:  (I have indicted my replies with a '>')

Reviewer # 1

1, I find the calculation of net radiation for all the meteorological stations does not really contribute to the paper. If I understand correctly, the measured net radiation is not really used except in validation against the measured ones at the energy balance stations. It is interesting as a stand-alone piece but is not well integrated into this paper. And I find removing this does not change the paper significantly or even make the paper more focused. One supporting evidence for my argument is that none of the related calculations

was even mentioned in the abstract and conclusions. Another reason I think this should be removed is that the calculations were not studied deeply. For example, I think the explanation of biases in the calculated net radiation is not thorough. The biases in the net longwave radiation in Figure 3 are so large that I don't think it is due to the wavelength of the instrument. Give these two reasons, I would strong suggest remove this part.

>This is understandable; it arose from a discussion among co-authors.  I moved this figure and the discussion surrounding it to the supplementary material.

Minor comments: 1, line 23, page 8: I would not use consistent. Maybe 'smooth'.
> Thank you for this suggestion. This has been changed.

2, line 22-25, page 9: This should go to a place much earlier when the radiative and flux measurements are first discussed (for example in Figure 4).
> This is a good point; it has been moved to the discussion of figure 5 on the previous page.

3, Figure 7 does not reveal much new information and should be removed. On the other hand, the discussion of Figure 8 is rather superficial at this moment. I only see 1 sentence for Figure 8 (line 20, page 9).
> More discussion has been added for figure 8.  Figure 7 and related discussion has been added to supplementary material.

4, Figure 10 (section 3.3.1): R2 should be provided in order to claim ?NDVI and soil moisture have the strongest positive correlation with evaporative fraction?.
>A table of R2 values has been added to the figure.

5, Figure 11 has too much non-essential and redundant information. The only helpful information is the middle panel showing the validation of the model. The others can be safely removed. On the other hand, Figure 12 can be combined with the middle panels of Figure 11. By the way, figure 12 is not even mentioned.
> The validation of the model and figure 12 has been combined and discussion has been developed around figure 12.

6, line 9, page 11: should be figure 9, not figure 8.
> Thank you. This has been changed.

Reviewer # 2

>  Based on AR2's recommendations, which were quite helpful, I have added some more development and discussion, in particular regarding additional work that has come out from the region.

The easiest way to deal with this point is probably to modify the abstract and develop some aspects of the physical basis of E fluxes, but I encourage the authors to develop

more thoroughly the NDVI-based model (although so far, the EF derived time series seems unrealistically high to me).

> As you recommended, I have made it clear that our fitting of evaporative fraction to NDVI and soil moisture is more of an exploration of the physical basis of evaporation and not a full-scale model, ready for large-scale implementation in the discussion.

> Your comments regarding the title are very helpful and I changed it to "Evaporation from Cultivated and Semi-Wild Sudanian Savanna in West Africa" because based on the floristic composition of the site, we are solidly in the biogeographic / phytosociological zone, and not the sahelian or Sudano-sahelian. Upon request, in future publications, or in supplementary materials, I could provide general botanical inventories that we completed early in our time on the site that support this classification. Semi - arid refers to the highly seasonal precipitation pattern.

L 20: I am not sure that you actually showed evidences that the fact that fluxes above the savanna-forest were higher was due to the number of rocks and trees and tree productivity.

> We did measure higher fluxes and explain it with the tree productivity.

L25: I think a paragraph is missing in the paper, as Figure 12 is not explained, and it is never written in the main text that NDVI only is sufficient to predict EF. This should be also further discussed.

> Thank you for bringing this to my attention. I have reconfigured some of the figures as per the recommendations of the first reviewer and I think that the connections between the figures and the text will be more apparent.

p2.L2: This is not straightforward: it should be sustained by a proper reference or moderated by adding 'for example'. For instance, there could be among-species differences in leaf renewal timing for some trees, not necessarily related to moisture availability.

> You are right. This sentence is meant to be rather general pointing out that there are some evergreen species on our site but most are deciduous. I changed the wording to make it clear that this is a general statement.

P2.L18-25: I would move these equations to section 2.3 (as is done with EF- or move EF equation to the introduction, although this relates more to 'measurements and calculations' than to an introduction).

> This is a good suggestion. I moved the equations to the Flux Calculation

P2.L26-32: consider removing some references. 2-3 references by statement are enough.

> While I think this is usually true, in this case, I want to demonstrate that it is an ubiquitous problem.

P2.L32: Let me suggest a few references that should be carefully studied and probably discussed at the end of the article, because they are located either in similar eco-climatic context, or bounding (either in the south or in the north) the study area. (Brummer et al., 2008; Guyot et al., 2009, 2012, Lohou et al., 2010, 2014, Mamadou et al., 2014, 2016; Velluet et al., 2014)

> Thank you for these references. I will add some discussion at the end of the article.

[Figure]

P3.L6: It is not the evaporative fraction, which is based on the concept of self-preservation, but the method that estimates daily evapotranspiration from evaporative fraction?

> Thank you, this is clarified.

P3.L15: This has been studied in West Africa and for a broad range of eco-climatic contexts by (Lohou et al., 2014)

> Thank you for the additional references. I will incorporate them appropriately.

P3.L19: is 'raised' plateau an appropriate term? I have never heard it.

> you are correct, the "raised" is redundant. I will remove this.

P3.L20: This is not true. I have not checked all the literature, but at least (Guyot et al., 2012; Lohou et al., 2014; Mamadou et al., 2016; Velluet et al., 2014) have studied energy fluxes in the area for different land covers and several seasonal cycles.

> Again, thank you for the additional references. I will incorporate them appropriately. P3. L24-L27: I would move that to the conclusion.

> Thank you for this recommendation.

P4. L 16-17: I am not sure this is relevant here.

> Since I do discuss social perception, I think it is relevant. The ethnic identity is the main way that the village identifies itself and is a primary determinant of land use.

P4. L18: It is not the sudanian area which is defined by alternating wet and dry season, but rather the monsoon climate. My suggestion would be (if you choose to keep sudanian instead of sahelo-sudanian): 'The watershed falls within the sudanian zone of the west-African monsoon climate system, with alternating dry and wet seasons with the rainfall falling. . ..'

> This is a helpful precision, Thank you.

P5.L7-8: This is an important remark: why not using ground T being recorded with the Decagon moisture probes for calculating G Only two depths are enough for a first calculation using the Harmonic method (Guyot et al., 2009). This would allow to be less elusive about the residuals of the energy budget, and to bring more evidence to conclude on higher/lower G under fields/Savanna.

> This is a good suggestion. I will add this in current or future publications.

P6.L6: I really think that individual times series should be shown, because according to Figure 5, they can be much different. At least one for the forest-Savanna cover and another for the field cover, and for the two depths, on a same figure containing H, LE and RN for the two land covers and for the same period. Maybe NDVI too?

> Thank you for the suggestion. I will reorganize some of the figures to have a figure that just has time series.

P6-7: section 2.8; I do agree with Reviewer #1 who proposes to remove this section. It could bring some interesting results, but would need much more developments, which are not the purpose of that paper. This would allow to gain some room for further developments in the text.

> Thank you for this remark, it is helpful to make a final decision.

P8.L10. To me, the NDVI for the forest in the dry season is not as much greener as you would expect from a forest dominated by Vitellaria paradoxa, which only need about a

month for leaf renewal. The NDVI pixel must include some significant herbaceous cover. If possible, could you estimate the vegetation classes distribution within the pixel?

> This is a good suggestion, it is clear that all of the pixels are very mixed. I will try to include a vegetation class estimation in this publication or in the planned publication about the spatial variation of evaporation. There is nowhere in the catchment that the forest is dominated by V. paradoxa. V. paradoxa is an agroforestry tree most often found in fields outside of this study area.

P8.L17. These are not average diurnal cycles, but single days samples. For me, all the results and conclusions drawn from this analyze (otherwise very interesting!) are weak due to the under sampling of diurnal cycles. For instance, we could not state from these single days, that 'Net radiation is slightly higher during the wet season (P8.L21)', (although this is probably true). If there is enough data to produce composite diurnal cycles over larger periods, please do. It is very hard to understand why there is such a significant LE flux in April over the field, which should be composed of bare soil according to the text and pictures. I can only suspect that there has been a rain in the previous day(s), - and there has been rain events in April according to Fig.4, and that the single day sample is too particular to be analyzed as a representative day. There may also be some other processes acting, such as lateral subsurface water transfer, and that should be discussed, but I doubt so. Another solution is to provide time series of RN, LE and H (and SM, rainfall, NDVI), as proposed on the comment P6.L6, on a larger plot than Figure 8, and not separated by wind sectors.These results should be discussed in the light of previous results for diurnal cycles obtained in the area (Guyot et al., 2012; Mamadou et al., 2014, 2016), for instance.

> Thank you for the suggestions for data presentation. It is clear that picking representative days can never capture the amount of variation visible. In a previous iteration of the paper, I did have a composite day, but we removed it because it concealed too much information. The scatter plot comparison of the components of the energy balance shows the information the comparison that would be visible on a composite plot, though not the diurnal variation. Figure 9 breaks up these comparisons by individual month. I will reconsider the choice of graphics once I make a time series plot as you suggest.

P8.L25: there is also, for both sites, this interesting feature in the late afternoon that LE> RN and even slightly positive during the night, as noted in (Mamadou et al., 2014, 2016), for instance.

> Thank you for identifying this comparison, I will discuss it more in my improved comparison to other sites.

P8.L31: It was lower over the field (if I am not mistaken).

P8.31: again, the expression 'for all months' does not hold when only two days are compared.

> According to figure 9, which is a composite comparison of all the months individually, shows that the ratio of Rn-ag to Rn-forest in April is less than 1, which means that it is lower over the field.

> Thank you for catching this mistake.

P9.L2: this is an interesting statement, but it needs to be further supported. For in- stance, (Mamadou et al., 2016), for a forested site in similar conditions, noted an early LE peak in the dry season, but also that Soil moisture changed very little from the morning to the

afternoon (<0.1mm). They concluded that it was probably a temperature limitation of stomatal opening.

> Thank you for pointing me to this work.

P9.L2-7: this is not supported by a figure. Again, I think that time series of the fluxes should be shown. Also, I think the discussion on limitations should consider here conductances, which can be affected by several processes, such as moisture limitations, but also shading, or stomatal resistances.

> Thank you for the suggestion, I will include this in my revision and consider how to best support it with a figure.

P9.L5. What is the 'absolute maximum'

> Thank you for pointing me to this imprecision and confusion. I will rewrite it.

P9.L11. I am not sure about this conclusion. I think that dynamic aspects should be taken into account, as fields and savanna will not necessarily have equal amounts of RN at the same time. I am not entirely sure on this, and I think that calculating G from temperature probes could help on that.

> This is also a good suggestion, I will qualify it and think about how to support it better (or refute it).

P9.L15-19. I think this is similar to what is found in (Guyot et al., 2009)

> Thank you for identifying this comparison. I'll explore it more in depth.

P9L20-28: This relates to the footprint concept. To my point of view, the footprint should be explicitly calculated for specific periods. But I would understand that it could significantly impact the paper in its current form. At least, wind sectors of Figure 8 should be more discussed; For instance, I do not clearly understand what is the difference between all the time series of, say, the Field panel, how have they been calculated? Also, I may be wrong, but I guess that N-W winds are the Harmattan winds, bringing hot, dry, and dusty air from the Sahara, and S-E winds would be the monsoon inflows, bringing moist air? If so, and as this has significant impact in the resulting energy budget, it could deserve some further insights (see e.g. (Guyot et al., 2012)).

> The wind sector analysis was a strategy to interpret the data based on the wind direction without a foot print analysis. Thank you for pointing out the aspects that are not clear, I will explain it better and further put it in context based on current literature.

P9.L25-27. I agree with Reviewer #1 that Figure 7 does not bring much, thank you for agreeing to move this figure to the appendix. Also, I am not sure that such correlation is an evidence for a causal effect.

P10.L2. Burning practice should be defined in the site description.

> I will. Thank you for the suggestion.

P10.L11-15: thank you for the correlation coefficient added on this figure.

P10.L25: My guess is that there are serious temporal limitations in this approach, and although the correlation coef are not too bad, the approach could probably be improved by considering different time windows. For instance, in the field, there is bare soil in the dry season, and for a little while, only soil evaporation may occur, and there is probably no need to take NDVI into account. On the opposite, once the herbaceous layer has grown, the system may not be water limited anymore, and soil moisture is not needed anymore. During the growing phase only, the equation could probably produce better results.

Also, what soil moisture time series have you used? If this is the catchment averaged one, then the higher regression slope on soil moisture for the field further indicate a higher

moisture-controlled system. If the ultimate goal is to compare it with remote sensing-derived soil moisture, this could easily be added here.

> These are very good suggestions. I would like to refine the model more, however perhaps it is beyond the scope of this paper? I will try to explain my soil moisture time series better in my revision. I did use a catchment averaged soil moisture, which does mean there is some bias, since there were more sensors and better working sensors, in the cultivated field.

P11.L4-10: some statements need to be made clearer (but maybe this is due to my poor english):

'sensible heat flux showed the greatest diff. in November & March': agree for November, although October seems even more different, but the ratio in March is close to 1. Unless the two mentioned surfaces are the two field surfaces (2009: millet, 2010 fallow)? But then it should be June and September. Again, for LE flux: the strong difference between forest and field are in March. I guess you point to the millet and fallow land use? Could you further discuss on why the fallow evapotranspirate less?

> Thank you for these comments, this is helpful to know where I need to go into more depth in my discussion. I'll improve these passages. It should be understandable even to a non-english speaker!

L7-9: I think again Figure 8 is not clear enough, why not having a plot ( as suggested above) with the fluxes of forest + field on a similar panel? There could be a panel of Rn, of LE, H, Soil moisture(+precip), and, say, NDVI?

> Thanks for the suggestion. I'll see how I can improve this figure and the time series in general. Perhaps your other suggested figures will also improve this point.

L9: The strongest difference could also be expected in the dry season, no? (see e.g. (Mamadou et al., 2016)) Or please expand on why this is expected.

> I would think this would be the highest difference because one site is still "moist" and the other is dry at this time. I'll see if Mamadou's findings help me explain this in a more relevant and convincing way.

On Rn: (Guyot et al., 2009) described a higher Rn on the dry season over woody cover, because surface temperature was lower, and LWnet higher. This could be discussed (if judged relevant), because there is an opposite behavior here. Although according to Fig.7 LWnet is higher in the forest in the dry season, and there are two tails in the LW-net box plot of Figure 6.

> I'll discuss it in contrast to Guyot's findings. This is very helpful.

P11.L15-16: I am not sure why rock presence should be mentioned here. Plus, they are not the only producers of H fluxes. Maybe an extra sentence could be added? And to clearly show that they are somehow responsible for a good part of H flux, a footprint analysis should be undertaken.Also, according to Figure 1, there is a strong topographic difference close to the forested area. I am not a specialist, but could that affect fluxes in any way?

> Yes you are right. This is why I performed an extra coordinate transform on the data, which goes beyond the normal procedure of data correction. I'll try to pull up some more examples of how rock presence changes the energy balance.

P11.L18: this is not clear to me: are we talking about trees getting water from open waters in the channel? If not, it should be simply stated that upper trees have 'probably' access to shallow groundwater at the origin of the springs.

> Yes, that is a smart way to word it.

P11.L20: what is permanent?

[Figure]

> The spring, I'll clarify.

P11.L20-21: I guess reference to the land cover is missing: this sentence relates to the field, right?

> yes you are right, I'll clarify

P11.L26. According to the picture in Figure 5, the field location is not particularly covered with more vegetation. The LE fluxes there are really high, and I have no explanation for this. On a Large Aperture scintillometer beam pathway, including very mixed cover, and including also a gallery forest preserved for similar reasons (Guyot et al., 2012) had much lower LE fluxes for instance. Same observation for (Brummer et al., 2008) in South West Burkina Faso, at a similar latitude, or (Mamadou et al., 2016) in a fully forested parkland in North Benin. This is very surprising, and I have no explanation.

> I'll look more closely at each of their cases and try to offer a more convincing explanation.

I am also curious, I have looked at the satellites images in google map, and the field location seems to appear in a middle of a much larger forest. What is at the origin of this specific deforestation? (this is not relevant in the review, only personal interest).

> The area all around this site is preserved as hunting reserves and Arly National Park. When people were forced to stop living in the areas as the became reserved, they were granted permission to live and farm only on this small "island" inside of the conserved area. I am sorry I didn't make this history more clear. As a side project during this research, I collected some oral histories of this transition. I hope to publish something from these interviews as well, although it will probably be in a different journal.

P12.L5-8, this should be rewritten according to the references given in the comment P2.L32. There is a site in Burkina Faso in similar climatic context, and sites in Benin are not that far south, and as for the cultivated ones, have probably a rather similar ecological context.

> Thank you again for pointing me to that work.

P12.L11-18: I have read that references to figure 12 have been added, as well as more discussion. Currently there is a strong issue that really needs to be addressed: The EF clearly shows that fluxes are very high, and this should be commented. EF could be compared with the study of (Lohou et al., 2014), which spans different eco-hydrological contexts through the N-S West-African gradient. In your Figure 12, EF does not seem to drop below 0.5, which is exceptionally high. If I roughly take a yearly average value of 0.6, and a yearly average daily value for ETo of say 5mm (Wang et al., 2007), which is probably a lower bound for ETo in this area, this produces about 1100mm or evapotranspiration yearly, which is much higher than precipitation. Footprint analysis could be undertaken to identify the reason for such high evapotranspiration. This could lead to very promising result.

> Yes, thank you for this precision. Perhaps I will add a table to directly compare our results to others to try to pull out reasons why ours are so different. I will think about adding a footprint analysis in the future.

P12.L23: cultural taboos, or health-related issues (malaria)?

[Figure]

UNIL | Université de Lausanne

> I am basing this on the best ethnography that I found of gourmantche agricultural practices. I did not study the full motivation behind historical taboos.

P13.L8: Add 'in the field', to make it clearer
> Thank you for this suggestion

Figures: In general, figure captions are too long and should be limited to description and not interpretation, but Figures are of good quality.
> Thank you, I will try to shorten them and put more in the text.
Figure 1: If possible, I would prefer the elevation shown as contour lines to better see the satellite image below. For instance, it is impossible to locate the village. And also, it is not easy to understand what is the dark line S-E / N-W just north of the study area: is it a plateau edge or a riparian area? I would recommend to put contour lines of the whole map, and draw the catchment contours with a proper shape.
Remove 'land locked country'? and 100m of elevation change. . .and the plateau?. If this is important, it should be put in the main text.
> Okay thank you for this, it is also in the main text, so I'll remove it here. I'll try to improve the map.

Figure 2: Figure caption: you could remove the last sentence, which is interpretation
> Okay, I will put it in the text.

Figure 4: soil moisture is exceptionally high in 2011 (known to be a dry year in the area, so I doubt there is any issue with the rain record), could you comment on that?
> This is a good point. I'll try to find an explanation. But if I can't, it is worth pointing out this inconsistency.
Figure 5: thank you for putting in the same figure both photographs and diurnal cycles, this makes a nice figure.
> You are welcome.
Figure 6: Figure caption: you could remove interpretation sentences. I also think there is some confusion in the middle-bottom two description.
> Okay, I'll clarify and move interpretation to the text.

P2.L1: change sudanian to sahelo-sudanian
> As stated, based on the floristic composition of the site, we are solidly in the biogeographic / phytosociological zone, and not the sahelian or Sudano-sahelian. I have added a botanical inventory to support this.

P2.L16: add 'in' before Eq.(1)
> Okay, Thank you.
P3.L7: [. . .]the diurnal cycle 'of' each [. . .]
> Okay, Thank you.

[Figure]

P4. L3: [. . .] were made in 'a' small [. . .]
> Okay, Thank you.
P4. L21 'in the' Guinean zone
> Okay, Thank you.
P6.L9. remove 'infrared'
> Okay, Thank you.
P7.L4. where 'smoothed'
> Thank you, this part was moved to the Supplementary Material
P8.L5. remove first occurrence of 'air'.
> Okay, Thank you.
P8.L21. for the both sites'
> Thank you
P8.L25. Heat instead of heath
> Thank you
P9.L12: remove 'below'
> Thank you
P9.L21: contributed 'to' higher fluxes'
> Thank you
P10.L24: be fitted'
P11.L16: at the end of the 'wet' season'
> Thank you
P12.L27: has protected 'it'?
> Thank you

Open - Reviewer # 3

Major Comment: Firstly, the paper is poorly organized, with much extensive content, though not exhaustive and too much interpretation in the results. While the methodology, in particular the eddy covariance data treatment requires a particular attention to have reliable turbulent fluxes, this was partially presented by the authors and the units of keys variables were omitted. In addition, there was a total confusion in the signification of such variables. For example, the available energy is not the sum of turbulent fluxes (H+LE) but rather the difference between the net radiation and the soil heat flux (Rn-G), see L13, L25, p5; and section 3.2.2.
> I have tried to make my wording and variable reference more precise and detailed.

Secondly, one of the main points of this paper was in the site comparison; however, basic information about the research sites was lacking. Did both sites have similar soil characteristics? The large differences in soil water content may indicate site differences in soil texture.
> I include more information about soil characteristics.

Also, more information is needed about the flux footprint. What was the fetch? Was the vegetation in particular (the rain fed site) within the flux footprint homogenous? The forest site seems to be located in a very complex topography according the map of the site (Figure 6). How this has been taken into account in the analysis of eddy covariance data?

[Figure]

UNIL | Université de Lausanne

> This was also a comment from AR2. We used a planar tilt correction to correct for the positive average wind speed as described by Oldroyd, H. J., Pardyjak, E. R., Huwald, H. and Parlange, M. B.: Adapting Tilt Corrections and the Governing Flow Equations for Steep, Fully Three-Dimensional, Mountainous Terrain, Bound.-Layer Meteorol., 1,Äì27, 2015.

> While I think a flux footprint would be interesting, I am saving it for a future analysis due to certain constraints that I discuss in my reply to AR2.

These aspects are important for understanding and interpreting the results. Finally, what is the value of the slope?
> I add a calculation of slope for the different wind directions around the EC stations.

Some of the writings throughout the text may be rewritten in more compact and yet concise style without losing the message they want to convey to the readers.
> This is a good comment, I hope that I improved the writing.

Some conclusions are drawn without the support of data.
> I hope that I have improved the writing of conclusions

Abstract L18-20: Which period of the year? I am very surprised with this result!
>
L18-22: I don't agree with this deduction. The presence of rocks and trees cannot, from my point of view, allow you to say that the soil heat flux is higher in the fields. You should take care with this assertion since you don't have any in situ measurements or direct calculation/estimation of this term of the energy balance to reinforce your conclusion.
>
Introduction L15-25: Give the units of all variables the first time that they are used.
>
L14-16: The cited reference Foken, 2008 'The energy balance closure problem: An overview' is not an appropriate reference. T. Foken has never worked on the link between global atmospheric processes and the land surface atmosphere interaction. Please provide an appropriate reference.
> Foken among the other cited references have all discussed the challenges of energy balance closure. Burba, 2013; Domingo et al., 2011; Farhadi, 2012; Federer et al., 2003; Foken, 2008; Foken et al., 2009; GUO et al., 2006; Katul and Parlange, 1992; Krishnan et al., 2012, 2012; Kustas et al., 1994; Parlange and Katul, 1992; Williams et al., 2012
L18: Write LeE is 'latent heat flux' instead of latent energy flux.
> Okay, Thank you.
L29: Replace Evaporation by Evapotranspiration since you were talking about vegetated surface.
> Brutsaert, 1982: evaporation is usually adequate to cover all processes of vaporization - transpiration from vegetation and direct evaporation from the soil are difficult to separate, water that is transpired is also evaporated through the stomata of plants.
L30 -33: It is true that in situ data of energy and water vapor fluxes were limited 'in the past', but today there are a lot of studies which have been conducted in this part of Africa

(Guyot et al, 2009; 2012; Mamadou et al., 2014 and 2016; Velluet et al., 2014; Timouk et al., 2009; Ramier et al., 2009).
>
L12-15, p3: THIS is not general but depends on the region; authors should specify the region in which this result has been obtained.
>
L1 27, section 2.1 there are too much information's which are from my point of view not really essential for the interpretation of fluxes. An example "the village is made up of a majority...". I cannot get the importance of this sentence and elsewhere in the section 2.1.
> Thank you for your perspective.  The ethnic groups of the village are relevant because a link is made to traditional land use practices.

L32 p4: Infrared gas analyzers. Open path or closed path? Need to be precized
> Thank you.
L1 p5: Replace eddy correlation by eddy covariance. What is the distance between the two studied sites?
> Okay Thank you.
L10: This is not true!! Sensible and latent heat fluxes cannot be measured at a half hour time step if you really used eddy covariance system to measure the fluctuation. How the sampled data have been then processed? What are the selection criteria? Given the complexity of these measurements, it is very shocking to see that certain details were not presented. Why do you use day light measurements (8am ' 4pm) for the comparison? Give the reason of this choice.
> We measured everything at 10 Hz but only calculated fluxes at a half hour time step.  All covariance measurements were calculated exactly as said with the corrections stated.   We did not use any software besides our own Matlab code that has been used for many studies and was modified for Burkina Faso.  The errors of eddy covariance are much higher at night and at dawn and dusk.
L20: Give the unit of different variables of Eq 4.
> ok
L25: Here instead of using the day light measurements as mentioned in L10, you preferred to use midday average. Why?
>Evaporative fraction is only computed once per day.  The logic is exactly the same, but here we calculated when there error increased.  There is a whole body of literature on this.
L9, section 2.6: What do you called the incoming shortwave infrared radiation? Is it the incoming shortwave radiation? If yes, make it clearer.
> ok

P7, Give the units of variables of Eq 9 to Eq12.
> ok
L5, p7: Longwave incoming or reflected radiation?
> I am not sure to what you are referring.

L8: Write Ts instead of T s
> I am not sure to what you are referring.

L5: I cannot get the meaning of this sentence.

[Figure]

UNIL | Université de Lausanne

> I am not sure to what you are referring.

L17: You started by saying that energy balance varied according to the month, i agree and now you compare a single day in April and in July. This is not coherent with the title of the section.
> Yes, I hope it is clearer now.

L23: Write the sensible heat flux
>okay, thank you.

L24-25: In the sentence, 'by July the latent heat has surpassed the sensible heath..'. How do you explain this fact? Replace heath by heat in the sentence.
> This can be explained by moisture availability.

L26-27: What was the magnitude of this residual in the morning? residual is lower over the savanna (which values?) and what about the agricultural field?
> Give magnitudes.

L30: What is the dust season? How was it objectively defined? Although the dust is something common over the region, it should affect also the measurement of the second site. How do you explain the fact that the net radiation was lower over the savanna-forest?
>This has been clarified, Net radiation was lower over the savanna – forest due to differences in albedo that were discussed.

L31-32: I am very surprised with these results. They are contrary to those obtained over the region. . .I would like to see the temporal evolution of H and LE based on half hourly data over the two studied sites.
> Okay I hope that I put them into more context and improved the figures to your liking.
L1-4, p9: I cannot get the meaning of the sentence ?The timing of the peaks of latent energy. . .the peak in the diurnal was after noon?. This sentence may be rewritten in more compact and yet concise way. Replace latent energy by latent heat flux and elsewhere in the paper.
> Okay thank you.
L6- 7: I would like to see the diurnal cycle of the available soil moisture
> I will go into more detail regarding soil moisture, for more detail on the soil texture, please see Ceperley, 2014.

Section 3.2.2: What is the general correlation?
> You are correct, this is not a technical term.
L25: It is normal since the reflected longwave radiation depends on surface temperature (Eq. 9)
> good
Section 3.2.3: it is very surprising that the savanna-forest contributed more sensible heat flux throughout the year than the agricultural land. The convection above the agricultural field should be more than that of the forest because the 'exposed area' and also the presence of vegetation over the forest which should limit this process.

[Figure]

UNIL | Université de Lausanne

> Yes, I was also surprised.

Section 3.3.1 Could you give the values of coefficient correlation and their associated p-values? It seems that wind speed is also correlated with EF. How have you identified the two dominant variables? How landscape moisture availability can be expressed as both NDVI and soil moisture? Higher levels of soil moisture? Which levels? What is the total net radiation? In the sentence ?Total net radiation does not show a strong influence, suggesting that this is not a radiation limited system?, I do not see the data which support this conclusion.

> I have now added correlations to the figure.  Wind speed is less.

L21, replace supposition by hypothesis.
>ok

Section 3.3.2: Give the values of coefficient correlation and their associated p-values of your fit.
>ok These were in the original figure 11.  I will now move them to the text.

Section 4.1, L13-16: I am not sure for these explanations. . . L19-20: What allow you to say that the level of water availability is permanent? Show then the water table in dry and wet season?
> I looked at it every day all year.  I have pictures.  The water level is explored more in depth in my colleages thesis.

L21: Replace latent heat by latent heat flux (and elsewhere in the paper)
>ok

L7-9, p12: I cannot get the meaning of the sentence In table 1: The Li-7500 measures both H2O and CO2 concentrations not 'HO concentration'. It seems that authors only provided the height of sensors above the agricultural field. What is the height of eddy covariance and additional measurements above the savanna-forest?
> Our licor was not calculated for CO2.  Thank you for catching the typo with water. The two stations were identical in terms of height from surface.

Figure 4: Environmental parameters at study site (which one?)
> Averaged across all  except when other wise noted.

Figure 5: Write in the title Diurnal cycle of the energy balance components. In the title of
> Okay Thank you

Figure 6 : H+LeE is not the total available energy!!!
> Understood though it is the sum of turbulent fluxes which is equal to total available energy.

Figures 11 and 12 are not cited in the paper.
> This has been improved.

    Sincerely,

                Dr. Natalie Ceperley, University of Lausanne

**Evaporation from Cultivated and Semi-Wild Sudanian Savanna in West Africa**

Natalie C. Ceperley[1,2], Theophile Mande[2], Nick van de Giesen[3], Scott Tyler[4], Hamma Yacouba[5], Marc B. Parlange[1,2]

[revised manuscript text omitted]

Natalie Ceperley 19.5.2017 14:30

African Continent

Legend
*Small Meteorological Stations*
△ Agriculture
▷ Forest
▽ Wetland
★ Eddy-Covariance Stations
— Ephemeral Stream
*Elevation (m)*
■ 213
■ 245
■ 277
■ 308
■ 340
● Springs

Unknown

**Figure 1: Map of Experimental Watershed. The site is located next to the village of Tambarga, in the southeastern corner of Burkina Faso, West Africa. Energy balance stations, small meteorological stations, including those near agroforestry trees, and hydrologic monitoring stations are shown. Springs are located at the source of the ephemeral stream, at the base of the rocky escarpment.**

Natalie Ceperley 6.6.2017 08:13

Natalie Ceperley 6.6.2017 08:11

Natalie Ceperley 6.6.2017 08:12

| Instrument | Measurement | Height/Depth | Number | Interval | Time Span |
|---|---|---|---|---|---|
| CSAT-3 Sonic Anemometer (Campbell Scientific, Logan, UT, USA) | 3D Wind Speed and Direction, Air Temperature | 2.2 m | 3 | 20 Hz, proc. 30 min. | May 2009 - October 2010 |
| Li- 7500 Infrared Gas Analyser (LICOR, Lincoln, NE, USA) | $H_2O$ Concentration | 2.2 m | 3 | 20 Hz, proc. 30 min. | May 2009 - October 2010 |
| CNR2 Radiometer (Kipp & Zonen, Delft, The Netherlands) | SW LW Radiation | 2.1 m | 2 | 1 min. | October 2009 - October 2010 |
| HMP450 (Campbell Scientific, Logan, UT, USA) with Radiation Shield | Air Temperature, Air Humidity | 2.25 m | 2 | 1 min. | May 2009 - October 2010 |
| Pluviometer 3029 (Précis Mécanique, Bezons, Cedex, France) | Precipitation | 1 m | 1 | 0.1 mm | May 2009 - January 2015 |
| Davis Instruments (Hayward, CA, USA) | Precipitation | .02 m | 12 | 1 min. | May 2009 - January 2015 |
| Davis Instruments (Hayward, CA, USA) | Shortwave Solar Radiation, Incoming | 1.8m | 12 | 1 min. | May 2009 - January 2015 |
| Infrared thermometer TN901 (Zytemp, Taiwan, R.O.C.) | Surface Temperature | 1.1 m | 12 | 1 min. | May 2009 - January 2015 |
| SHT7 (Sensiron AG, Staefa ZH, Switzerland) | Air Temperature & Humidity | 1.7 m | 12 | 1 min. | May 2009 - January 2015 |
| 5TM, 5TE, ECTM (Decagon, Pullman, WA, USA) | Soil Humidity | 5-30 cm | ~ 24 (varied) | 1 min. | May 2009 - January 2015 |

**Table 1: Inventory of Instruments used for Energy Balance Analysis: The name of the sensor or instrument is followed by the measurement it performs, the height and depth of each sensor, the total number used, and the interval of measurement. The heights were identical at both measurement points.**

Natalie Ceperley 8.6.2017 15:34

Natalie Ceperley 8.6.2017 15:35

**Inventory of woody vegetation in Forest / Savanna ( 1 ha.)**

| | | | |
|---|---|---|---|
| Acacia macrostachya | Leg.-Mim. | mph | S |
| Burkea africana | Leg.-Caes. | mph | SZ |
| Combretum nigricans | Combretaceae | mph | S |
| Daniella oliveria | Leg.-Caes. | MPh | SZ |
| Detarium microcarpum | Leg.-Caes. | mph | S |
| Gardenia erubecens | Rubiaceae | nph | S |
| Grewia flavescens | Tiliaceae | Lmph | GC |
| Guiera senegalensis | Combretaceae | nph | SZ |
| Hymenocardia acida | Euphorbiaciaceae | mph | SZ |
| Lannea acida | Anacardiaceae | mPh | S |
| Parkia biglobosa | Leg.-Mim. | mPh | S |
| Prosopis africana | Leg.-Mim. | mPh | S |
| Pteleopsis subrosa | Combretaceae | mph | SZ |
| Pterocarpus erinaceus | Leg.-Pap. | mPh | S |
| Sclerocarya birria | Anacardiaceae | mph | S |
| Sterculia setigera | Sterculiaceae | mph | S |
| Strychnos spinosa | Loganiaceae | LmPh | PAL |
| Terminalia avicennioides | Combretaceae | mph | S |
| Terminalia laxiflora | Combretaceae | mPh | S |
| Terminalia schimperiana  (glaucenscens) | Combretaceae | mph | S |
| Terminalia mollis | Combretaceae | mph | PRA |
| Vitellaria paradoxa | Sapotaceae | mPh | S |
| Ximenia americana | Olacaceae | nph | Pt |

**Inventory of woody vegetation in Agricultural Fields**

| | | | |
|---|---|---|---|
| Acacia sieberiana | Leg.-Mim. | mph | SZ |
| Bombax costatum | Bobacaceae | mph | S |
| Detarium microcarpum | Leg.-Caes. | mph | S |
| Ficus sp. | Moraceae | MPh | |
| Lannea sp. | Anacardiaceae | mPh | S / SZ |
| Piliostigma reticulatum | Leg.-Caes. | mph | SG |
| Sclerocarya birrea | Anacardiaceae | mph | S |
| Terminalia schimperiana  (glaucenscens) | Combretaceae | mph | S |
| Terminalia laxiflora | Combretaceae | mPh | S |
| Terminalia mollis | Combretaceae | mph | PRA |
| Ziziphrus mauritaniana | Thamnaeceae | mph | PAL |

**Chorology**     **No.**

| | | |
|---|---|---|
| S | Sudanian | 19 |
| SZ | Sudano-Zambezian | 6 |
| GC | Guieo-Congolian | 1 |
| PAL | Paleotropical | 1 |

| | | | |
|---|---|---|---|
| Pt | Pantropical | 1 | – |
| SG | Sudano/Guinean transition | 1 | – |
| PRA | Pluriregional African | 1 | – |

| **Abbreviations** | | – | – |
|---|---|---|---|
| Leg.-Caes. | Leguminosae.-Caesalpinioideae | | |
| Leg.-Mim. | Leguminosae-Mimosoideae | | |
| Leg.-Pap. | Leguminosae-Papilionoideae | | |

| **Life Forms** | | – | – |
|---|---|---|---|
| mph | microphanerophyte | 2 - 8 m | – |
| MPh | megaphanerophyte | > 30 m | – |
| nph | nanophanerophyte | 0.5-2 m | – |
| L | Liana | | – |
| mPh | mesophanerophyte | 8-30 m | – |

**Table 2: Inventory of species found in both agricultural field site and forest savanna site.**

[Figure]

**Figure 2: Environmental Parameters at Study Site for Monitoring Period (2009 - 2015): a) Cloud cover measured with shortwave radiometers and averaged over all stations; b) Average NDVI from MODIS satellite data (250 meter resolution, 10-day composite) averaged over each landcover of pixels containing stations; c) Monthly average air temperature (red line), also averaged for all stations; d) Volumetric soil moisture averaged over all stations and all land covers between 5 and 30 cm depth (blue line); And e) monthly rainfall (missing bars indicate lack of rain).**

Natalie Ceperley 2.2.2017 19:41

[Figure]

|  | April 2010 | July 2010 |  |
|---|---|---|---|
| Agricultural Field |  |  | Agricultural Field |
| Savanna- Forest |  |  | Savanna-Forest |

**Figure 3: Diurnal Cycle of the Energy Balance.** The upper four photographs correspond to the four subplots - the date of the photograph is the same month as the representative plot of diurnal energy budget. Note that April (left) is the dry season and the atmosphere was very hazy, in part due to fires. July is the start of the rainy or wet season (right). The energy budget is made up of the sensible heat (H, blue), latent heat (LeE, green), net radiation ($R_n$, red) on the y-axis (W/m2) over the savanna-forest (middle row) and agricultural land (top row) according to time of day (x-axis, 24 hours). The residual of the energy budget is also shown in turquoise. The final row shows the half hour calculation of evaporative fraction. Daily averages were taken between 10 and 14 hours, shown with the vertical lines, with the savanna-forest in green and agriculture in blue.

Natalie Ceperley 22.5.2017 12:21

Natalie Ceperley 2.2.2017 19:41

[Figure]

**Figure 4: Time Series of Average Daily Sensible Heat Flux (1), Average Daily Net Radiation (3), Average Daily Residual (4), Volumetric Soil Moisture in the Agricultural Field (5), Volumetric Soil Moisture in the Savanna Forest (6), Daily Rain and NDVI (7). Plots 1 – 4 show the data from the energy balance stations over Agricultural Field (blue circles) and Savanna Forest (red crosses) in Watts per square meter. Plots 5 – 6 show the Volumetric Soil Moisture at 5 different depths over the same time period. Finally, plot 7 shows the NDVI for the pixels containing the energy balance station in the field (blue dashed) and in the savanna forest (red, solid). The rain in mm is shown as a bar graph.**

Figure 5: Comparison between fluxes measured over the savanna-forest and the field. In each plot, colour indicates soil volumetric water content, red is dry and blue is wet. The least square regression lines are shown in green and the 1:1 lines are in black. Measurements over the savanna-forest are on the x-axis and those over agriculture are on the y-axis. All fits were significant (p<0.005). The middle column shows the components of net radiation calculated with the net radiometers. The right shows the components of net radiation calculated using parameters measured at the small meteorological stations: Ta, incoming SW, Ts, etc.

[Figure]

[Figure]

**Figure 6: Two dominant wind sectors for each eddy-covariance set up are plotted: Above the two over the savanna field and below the 4 over the agricultural field. Mean latent heat flux for each month with standard deviations is shown in blue and sensible heat in green. Note that for some months there was no data.**

Natalie Ceperley 2.2.2017 19:43

... [209]

Unknown

Natalie 3.7.2017 21:57

Natalie Ceperley 2.2.2017 19:41

Natalie Ceperley 8.6.2017 15:05

Natalie Ceperley 3.7.2017 23:21

Natalie Ceperley 3.7.2017 23:21

[Figure]

Figure 7: **Examination of ratio of the daily fluxes by month and year. The ratios between flux measurements over the field and savanna-forest is shown according to month (x - axis) and year (in colour: blue is 2009 and red is 2010). The soil moisture value is shown with shading. Error bars show the standard deviation around the average of daily ratios. Points below the 1 line indicate when the savanna-forest flux is higher than the agricultural field flux. It is important to note that the field was farmed in 2009 until the end of July but left fallow in 2010.**

Natalie 3.7.2017 21:57

Natalie Ceperley 2.2.2017 19:41

[Figure]

| | Evaporative Fraction | |
|---|---|---|
| | Agriculture | Savanna-Forest |
| Vegetation (NDVI) | 69.76 | 74.71 |
| Soil Water content | 64.63 | 48.33 |
| Net Radiation | -6.88 | -13.33 |
| Wind Speed | -51.85 | -43.07 |
| Cloud Cover | 19.17 | 38.94 |

[Figure]

Natalie 3.7.2017 21:57

**Figure 8: Daily evaporative fraction over study period for the agricultural field (blue) and the savanna-forest (green) compared with the observed a) NDVI, b) VWC, c) wind speed, d) net radiation, and e) cloud cover. In all cases, environmental variables are from the average of stations with the same landcover with the energy balance station. These plots show a better correlation with NDVI and VWC (top row), suggesting moisture, not radiation, limited system. The least squared regression lines are shown for each plot. Correlations between variables are in the table.**

[Figure]

[Figure]

Natalie Ceperley 2.2.2017 19:41

Natalie Ceperley 3.2.2017 14:35

Natalie Ceperley 2.2.2017 19:45
Moved (insertion) [2]

Unknown

Natalie Ceperley 3.2.2017 14:35

Unknown

**Figure 9: Further Examination of relationship between Soil Moisture, Vegetation Index, and Evaporative Fraction. Upper plots show** quality of fit **of linear regression model relating soil volumetric water content, vegetation index, and evaporative fraction over agriculture (left) and over savanna-forest (right). The 1:1 line is in red and the 95% confidence interval is shown with dotted lines for the range of available soil and vegetation.** Lower plots shows evaporative fraction over the field (above) and the savanna-forest (below). Measured data points are in black (points), calculated evaporative fraction based on soil moisture and NDVI is shown in blue and NDVI alone is shown in green.

Natalie 3.7.2017 21:58

Natalie Ceperley 2.2.2017 19:44

Natalie Ceperley 2.2.2017 19:46

Natalie Ceperley 2.2.2017 19:46

Natalie Ceperley 2.2.2017 19:46

Natalie Ceperley 2.2.2017 19:46

Natalie Ceperley 2.2.2017 19:47

Natalie Ceperley 2.2.2017 19:47

Natalie Ceperley 2.2.2017 19:48

Natalie Ceperley 2.2.2017 19:45
**Moved (insertion) [1]**

Natalie Ceperley 2.2.2017 19:48

Natalie Ceperley 2.2.2017 19:48

Natalie Ceperley 2.2.2017 19:49

Natalie Ceperley 2.2.2017 19:45
**Moved up [2]:** ... [210]

Unknown

Unknown

Natalie Ceperley 2.2.2017 19:45

Natalie Ceperley 2.2.2017 19:45
**Moved up [1]: Time series of evapora** ... [211]

**Supplementary Material**

**S1.1 Net Radiation**

To account for any scale discrepancies between the small meteorological stations and the energy balance stations, net radiation was calculated at each small station (Brutsaert, 1982). Net shortwave radiation was calculated using the measured incoming shortwave radiation (table 1) and albedo (α) Eq. (S1):

$$sw_n = (1 - \alpha) \cdot sw_i. \text{ (S1)}$$

Albedo was measured using solar radiometers facing up and down for 155 days between October 2011 and November 2012. Since albedo is heavily correlated with soil moisture and vegetation cover, a linear regression model of albedo based on soil moisture and NDVI was used to estimate albedo when it was not measured. When either soil moisture or NDVI was not available, a linear model based just on the other, available metric was used. Gaps were filled with a linear interpolation and data missing at the beginning of the observation were filled using a linear interpolation based on average albedo for the day of the year (between April 22-25, 2009). All estimated albedo values were smoothed using a 10-day moving average filter. Albedo fell within the acceptable range for the vegetation covers (Figure S1). Long wave radiation was calculated as the sum of long wave upwelling radiation, Eq. (S2):

$$Rl_u = \varepsilon_s \sigma T_s^4. \text{ (S2)}$$

where $\varepsilon_s$ is the surface emissivity, taken to have an average value (0.97), σ is the Stefan Boltzmann constant, and $T_s$ is the measured surface temperature, and the incoming long wave radiation, which is taken as a fraction of clear sky incoming long wave radiation, Eq. (S3):

$$Rl_i = Rl_{ic}(1 + am_c^b), \text{ (S3)}$$

where Eq. (S4):

$$Rl_{ic} = \varepsilon_{ac} \sigma T_a^4. \text{ (S4)}$$

and $m_c$ is the measured cloud cover (6), a and b are constants, $\varepsilon_{ac}$ is the atmospheric emissivity during clear sky conditions, and $T_a$ is the measured air temperature. The atmospheric emissivity is Eq. (S5):

$$\varepsilon_{ac} = a'(\frac{e_a}{T_a})^{\frac{1}{7}}. \text{ (S5)}$$

where $e_a$ is the vapor pressure near the surface, determined with the measured relative humidity and air temperature and a' is calculated with a Beta function and air temperature and averages 1.24 at our site as it does for average meteorological conditions (Brutsaert, 1982). Comparison to the measured net radiation at the energy balance station (figure S2) shows some discrepancies. These may be due to the difference in wavelengths measured by the solar radiation sensors at the small meteorological stations, which use a silicon photodiode detector to detect radiation at wavelengths of 300 to 1100 nanometers. Whereas the pyranometer measures from 300 to 2800 nm and the pyrogeometer from 4.5 to 42 μm. The spectral response of the individual sensors was not available at this time, but estimation using standard spectra (ASTM G173-03 Reference Spectra) integrated over the two different short wave ranges indicates that the solar radiation sensors measure

Natalie 3.7.2017 12:20

Natalie 3.7.2017 12:20

61.78% of the energy that the pyranometers measure. These differences would be further exaggerated by the geometry of the sensor ($180^o$ for $sw_i$ and $150^o$ for $sw_u$ from the pyranometer versus the solar radiation sensor which is less than 100% for the full $180^o$) especially in dusk and dawn conditions. Additionally, the effect of albedo varies according to cloudiness and sun altitude. Most of the radiation emitted by the earth and atmosphere is between 4 and 100 µm and measurements are often flawed because instruments themselves emit radiation of comparable wavelengths and intensity to the long wave radiation that we want to measure. Thus our comparison is reasonable because it is of a similar order of magnitude, further fine-tuning the calculation will be done in a subsequent work.

**References**

Brutsaert, W.: Evaporation into the Atmosphere: Theory, History, and Applications, Kluwer Academic Publishers, Dordrecht, The Netherlands., 1982.

[Figure]

**Figure S1: Distribution of albedo by landcover (Agricultural, Wetland, and Forest) and by month for the study period. The central dot is the median, the edges of the box are the 25th and 75th percentiles, the whiskers extend to the most extreme data points not considered outliers, and outliers are plotted individually. Average values for various land-covers listed by Brutsaert (1982) are drawn and labelled to the right. Ranges for these are shown in dotted lines. Our values mostly range between those of bare field or green grass and dry soils and desert, with dry prairie or savanna making a good mid value.**

Unknown

[Figure]

| | Agriculture | Savanna-Forest |
|---|---|---|
| a) LW$_{net}$ | 0.58 | 0.50 |
| b) SW$_{net}$ | 0.76 | 0.78 |
| c) $R_n$ | 0.80 | 0.77 |
| $R_n$-H- L$_e$E | 0.86 | 0.86 |

**Figure S2: Comparison of measurements and calculations on a half hour scale of (a) net short wave, (b) net long wave, and (c) total radiation and the residual (not shown, correlation in table) after subtracting the measured turbulent fluxes. In all cases the slope is less than 1, which means that the measurements are greater than the calculation. The slopes of the regression lines are in the accompanying table. In each plot, measurements over the agricultural field are shown in blue and over the savanna-forest are shown in green. Black lines show the one to one line and coloured lines show the least squared regression lines.**

[Figure]

**Figure S3: Net Long-wave Radiation Response to Soil Moisture and Surface Temperature.** Surface temperature and net long wave radiation had a strong negative correlation with each other and also with volumetric soil water content, shown by the colour. The pattern of this correlation varied by land cover, savanna-forest above and agriculture below.

Unknown

| Page 2: [1] Deleted | Microsoft Office User | 20.02.17 17:21 |
|---|---|---|

the seasonal cycle of

| Page 2: [1] Deleted | Microsoft Office User | 20.02.17 17:21 |
|---|---|---|

the seasonal cycle of

| Page 2: [1] Deleted | Microsoft Office User | 20.02.17 17:21 |
|---|---|---|

the seasonal cycle of

| Page 2: [2] Change | Unknown | |
|---|---|---|

Field Code Changed

| Page 2: [3] Change | Unknown | |
|---|---|---|

Field Code Changed

| Page 2: [4] Deleted | Natalie | 03.07.17 12:19 |
|---|---|---|

(Abiodun et al., 2008; Feddema et al., 2005; Mande et al., 2011; Steiner et al., 2009; Sylla et al., 2015; Vitousek, 1997)(Abiodun et al., 2008; Feddema et al., 2005; Mande et al., 2011; Steiner et al., 2009; Sylla et al., 2015; Vitousek, 1997)

| Page 2: [5] Change | Unknown | |
|---|---|---|

Field Code Changed

| Page 2: [6] Change | Unknown | |
|---|---|---|

Field Code Changed

| Page 2: [7] Deleted | Natalie | 03.07.17 12:19 |
|---|---|---|

(Foken, 2008; Szilagyi and Parlange, 1999)(Foken, 2008; Szilagyi and Parlange, 1999)

| Page 2: [8] Moved to page 5 (Move #3) | Microsoft Office User | 20.02.17 17:24 |
|---|---|---|

 The surface energy budget is written Eq. (1):

$R_n = L_e E + H + G,$ (1)

where $L_eE$ is latent energy flux, $H$ is sensible heat flux, $R_n$ is the net radiation, and $G$ is the soil heat flux. The sensible heat, Eq. (2):

$H = \rho c_p \overline{w'T'},$ (2)

and latent heat, Eq. (3):

$L_e = L_e \rho \overline{w'q'},$ (3)

fluxes can be obtained from eddy covariance and the using the above equations, where $\rho$ is the air density, $c_p$ is the specific heat, $\overline{w'T'}$ is the covariance of fluctuations of vertical wind speed and temperature, $L_e$ is the latent energy of vaporization, and $\overline{w'q'}$ is covariance of fluctuations of vertical wind speed and humidity.

| Page 2: [9] Deleted | Natalie Ceperley | 28.06.17 12:52 |
|---|---|---|

 (1):

$R_n = L_e E + H + G,$ (1)

where $L_e E$ is latent energy flux, $H$ is sensible heat flux, $R_n$ is the net radiation, and $G$ is the soil heat flux.

The sensible heat, Eq. (2):

$H = \rho c_p \overline{w'T'},$ (2)

and latent heat, Eq. (3):

$$L_e = L_e \rho \overline{w'q'}$$

| Page 2: [9] Deleted | Natalie Ceperley | 28.06.17 12:52 |
|---|---|---|

(1):

$R_n = L_e E + H + G,$ (1)

where $L_e E$ is latent energy flux, $H$ is sensible heat flux, $R_n$ is the net radiation, and $G$ is the soil heat flux.

The sensible heat, Eq. (2):

$H = \rho c_p \overline{w'T'},$ (2)

and latent heat, Eq. (3):

$$L_e = L_e \rho \overline{w'q'}$$

| Page 2: [10] Change | Unknown | |
|---|---|---|

Field Code Changed

| Page 2: [11] Deleted | Natalie Ceperley | 04.07.17 02:27 |
|---|---|---|

(Burba, 2013; Domingo et al., 2011; Farhadi, 2012; Federer et al., 2003; Foken et al., 2009; GUO et al., 2006; Katul and Parlange, 1992; Krishnan et al., 2012, 2012; Kustas et al., 1994; Parlange and Katul, 1992; Williams et al., 2012)(Burba, 2013; Domingo et al., 2011; Farhadi, 2012; Federer et al., 2003; Foken et al., 2009; GUO et al., 2006; Katul and Parlange, 1992; Krishnan et al., 2012, 2012; Kustas et al., 1994; Parlange and Katul, 1992; Williams et al., 2012)

| Page 2: [12] Change | Unknown | |
|---|---|---|

Field Code Changed

| Page 2: [13] Deleted | Natalie | 03.07.17 12:19 |
|---|---|---|

(Brutsaert, 1982; Brutsaert and Parlange, 1992; Burba, 2005; Compaore, 2006; Crago, 1996; Crago and Qualls, 2013)(Brutsaert, 1982; Brutsaert and Parlange, 1992; Burba, 2005; Compaore, 2006; Crago, 1996; Crago and Qualls, 2013)

| Page 2: [14] Change | Unknown | |
|---|---|---|

Field Code Changed

| Page 2: [15] Deleted | Natalie | 03.07.17 12:19 |
|---|---|---|

(Bagayoko et al., 2007; Dolman et al., 1997; Gash et al., 1997; Mande et al., 2011)(Bagayoko et al., 2007; Dolman et al., 1997; Gash et al., 1997; Mande et al., 2011)

| Page 2: [16] Change | Unknown | |
|---|---|---|

Field Code Changed

| Page 2: [17] Deleted | Natalie | 03.07.17 12:19 |

(Charney, 1975; Nicholson et al., 1998)(Charney, 1975; Nicholson et al., 1998)

| Page 2: [18] Change | Unknown | |

Field Code Changed

| Page 2: [19] Deleted | Natalie | 03.07.17 12:19 |

(Brutsaert and Sugita, 1992; Compaore, 2006; Porte-Agel et al., 2000; Shuttleworth, 1989; Szilagyi et al., 1998; Szilagyi and Parlange, 1999)(Brutsaert and Sugita, 1992; Compaore, 2006; Porte-Agel et al., 2000; Shuttleworth, 1989; Szilagyi et al., 1998; Szilagyi and Parlange, 1999)

| Page 2: [20] Change | Unknown | |

Field Code Changed

| Page 2: [21] Deleted | Natalie | 03.07.17 12:19 |

(Brutsaert and Sugita, 1992; Porte-Agel et al., 2000)(Brutsaert and Sugita, 1992; Porte-Agel et al., 2000)

| Page 3: [22] Change | Unknown | |

Field Code Changed

| Page 3: [23] Deleted | Natalie | 03.07.17 12:19 |

(Bateni and Entekhabi, 2012)(Bateni and Entekhabi, 2012)

| Page 3: [24] Change | Unknown | |

Field Code Changed

| Page 3: [25] Deleted | Natalie | 03.07.17 12:19 |

(Gentine et al., 2007)(Gentine et al., 2007)

| Page 3: [26] Change | Unknown | |

Field Code Changed

| Page 3: [27] Deleted | Natalie | 03.07.17 12:19 |

(Lhomme and Elguero, 1999)(Lhomme and Elguero, 1999)

| Page 3: [28] Change | Unknown | |

Field Code Changed

| Page 3: [29] Deleted | Natalie | 03.07.17 12:19 |

(Farah et al., 2004; Lohou et al., 2010, 2014)

| Page 3: [30] Deleted | Natalie Ceperley | 28.06.17 13:30 |

(Farah et al., 2004).

| Page 3: [30] Deleted | Natalie Ceperley | 28.06.17 13:30 |

(Farah et al., 2004).

**Page 3: [31] Deleted**           **Natalie Ceperley**           **23.05.17 11:03**

makes

**Page 3: [31] Deleted**           **Natalie Ceperley**           **23.05.17 11:03**

makes

**Page 3: [32] Change**           **Unknown**

Field Code Changed

**Page 3: [33] Deleted**           **Natalie Ceperley**           **04.07.17 02:13**

(Bagayoko et al., 2007; Ezzahar et al., 2009; Guyot et al., 2009; Mauder et al., 2006)(Bagayoko et al., 2007; Ezzahar et al., 2009; Guyot et al., 2009; Mauder et al., 2006)

**Page 3: [34] Change**           **Unknown**

Field Code Changed

**Page 3: [35] Deleted**           **Natalie**           **03.07.17 12:19**

(Nadeau et al., 2009; Simoni et al., 2011)(Nadeau et al., 2009; Simoni et al., 2011)

**Page 3: [36] Moved to page 14 (Move #4)  Microsoft Office User**           **22.02.17 16:26**

Our results point to the necessity for ground measurements for eventual up-scaling from point to regional evaporation measurements in remote and less-studied regions of the globe. We began this work with discussion with community partners and to bring it full circle we conclude this paper by relating it back to the cultural context.

**Page 3: [37] Change**           **Unknown**

Field Code Changed

**Page 5: [38] Moved from page 2 (Move #3)Microsoft Office User**           **20.02.17 17:24**

The surface energy budget is written in Eq. (1):

$$R_n = L_e E + H + G, \text{ (1)}$$

where $L_e E$ is latent energyheat flux, $H$ is sensible heat flux, $R_n$ is the net radiation, and $G$ is the soil heat flux, all in Watts per square meter (W m$^{-2}$).

The sensible heat, is expressed in Eq. (2):

$$H = \rho c_p \overline{w'T'}, \text{ (2)}$$

where $\rho$ is the air density (kg m$^{-3}$), $c_p$ is the specific heat (J kg$^{-1}$ K$^{-1}$), $w'T'$ is the covariance of fluctuations of vertical wind speed (m s$^{-1}$) and temperature (K). Land latent heat flux, is expressed in Eq. (3):

$$L_e = L_e \rho \overline{w'q'}, \text{ (3)}$$

fluxes can be obtained from eddy covariance and the using the above equations, where $\rho$ is the air density, $c_p$ is the specific heat, $w'T'$ is the covariance of fluctuations of vertical wind speed and temperature, $L_e$ is the latent energy of vaporization (J g$^{-1}$), and $w'q'$ is covariance of fluctuations of vertical wind speed (m s$^{-1}$) and humidity (g m$^{-3}$).

| Page 5: [39] Formatted | Natalie Ceperley | 28.06.17 13:14 |

Superscript

| Page 5: [40] Formatted | Natalie Ceperley | 28.06.17 13:14 |

Superscript

| Page 5: [41] Formatted | Natalie Ceperley | 28.06.17 13:15 |

Superscript

| Page 5: [42] Formatted | Natalie Ceperley | 28.06.17 13:15 |

Superscript

| Page 5: [43] Formatted | Natalie Ceperley | 28.06.17 13:15 |

Superscript

| Page 5: [44] Deleted | Natalie Ceperley | 28.06.17 13:09 |

fluxes can be obtained from eddy covariance and the using the above equations, where $\rho$ is the air density, $c_p$ is the specific heat, $w'T'$ is the covariance of fluctuations of vertical wind speed and temperature,

| Page 5: [45] Formatted | Natalie Ceperley | 28.06.17 13:15 |

Superscript

| Page 5: [46] Formatted | Natalie Ceperley | 28.06.17 13:15 |

Superscript

| Page 5: [47] Formatted | Natalie Ceperley | 28.06.17 13:15 |

Superscript

| Page 5: [48] Change | Unknown | |

Field Code Changed

| Page 5: [49] Deleted | Natalie | 03.07.17 12:19 |

(Brutsaert, 1982; Higgins, 2012)(Brutsaert, 1982; Higgins, 2012)

| Page 5: [50] Change | Unknown | |

Field Code Changed

| Page 5: [51] Deleted | Natalie | 03.07.17 12:19 |

(Aubinet et al., 2012; Burba, 2005; Oldroyd et al., 2015; Rebmann et al., 2012; Wilczak et al., 2001)(Aubinet et al., 2012; Burba, 2005; Oldroyd et al., 2015; Rebmann et al., 2012; Wilczak et al., 2001)

| Page 5: [52] Change | Unknown | |

Field Code Changed

| Page 5: [53] Deleted | Natalie | 03.07.17 12:20 |

(Foken et al., 2012; Leuning, 2007; Webb et al., 1980)(Foken et al., 2012; Leuning, 2007; Webb et al., 1980)

**2.8 Net Radiation**

To account for any scale discrepancies between the small meteorological stations and the energy balance stations, net radiation was calculated at each small station (Brutsaert, 1982). Net shortwave radiation was calculated using the measured incoming shortwave radiation (table 1) and albedo (α) Eq. (8):

$$sw_n = (1 - \alpha) \cdot sw_i. \ (8)$$

Albedo was measured using solar radiometers facing up and down for 155 days between October 2011 and November 2012. Since albedo is heavily correlated with soil moisture and vegetation cover, a linear regression model of albedo based on soil moisture and NDVI was used to estimate albedo when it was not measured. When either soil moisture or NDVI was not available, a linear model based just on the other, available metric was used. Gaps were filled with a linear interpolation and data missing at the beginning of the observation were filled using a linear interpolation based on average albedo for the day of the year (between April 22-25, 2009). All estimate albedo values were smooth using a 10-day moving average filter. Albedo fell within the acceptable range for the vegetation covers (Figure 2). Long wave radiation was calculated as the sum of long wave upwelling radiation, Eq. (9):

$$Rl_u = \varepsilon_s \sigma T_s^4 \ . \ (9)$$

where $\varepsilon_s$ is the surface emissivity, taken to have an average value (0.97), σ is the Stefan Boltzmann constant, and $T_s$ is the measured surface temperature, and the incoming long wave radiation, which is taken as a fraction of clear sky incoming long wave radiation, Eq. (10):

$$Rl_i = Rl_{ic}(1 + am_c^b), \ (10)$$

where Eq. (11):

$$Rl_{ic} = \varepsilon_{ac} \sigma T_a^4 \ . \ (11)$$

and $m_c$ is the measured cloud cover (6), a and b are constants, $\varepsilon_{ac}$ is the atmospheric emissivity during clear sky conditions, and $T_a$ is the measured air temperature. The atmospheric emissivity is Eq. (12):

$$\varepsilon_{ac} = a'(\frac{e_a}{T_a})^{\frac{1}{7}} \ . \ (12)$$

where $e_a$ is the vapor pressure near the surface, determined with the measured relative humidity and air temperature and a' is calculated with a Beta function and air temperature and averages 1.24 at our site as it does for average meteorological conditions (Brutsaert, 1982). Comparison to the measured net radiation at the energy balance station (figure 3) shows some discrepancies. These may be due to the difference in wave lengths measured by the solar radiation sensors at the small meteorological stations, which use a silicon photodiode detector to detect radiation at wavelengths of 300 to 1100 nanometers. Whereas the pyranometer measures from 300 to 2800 nm and the pyrogeometer from 4.5 to 42 μm. The spectral response of the individual sensors was not available at this time, but estimation using standard spectra (ASTM G173-03 Reference Spectra) integrated over the two different short wave ranges indicates that the solar radiation sensors measure 61.78% of the energy that the pyranometers measure. These differences

would be further exaggerated by the geometry of the sensor ($180^o$ for $sw_i$ and $150^o$ for $sw_u$ from the pyranometer versus the solar radiation sensor which is less than 100% for the full $180^o$) especially in dusk and dawn conditions. Additionally, the effect of albedo varies according to cloudiness and sun altitude. Most of the radiation emitted by the earth and atmosphere is between 4 and 100 μm and measurements are often flawed because instruments themselves emit radiation of comparable wavelengths and intensity to the long wave radiation that we want to measure. Thus our comparison is reasonable because it is of a similar order of magnitude, further fine-tuning the calculation will be done in a subsequent work.

| Page 8: [55] Deleted | Natalie Ceperley | 22.05.17 15:35 |
|---|---|---|

the

| Page 8: [55] Deleted | Natalie Ceperley | 22.05.17 15:35 |
|---|---|---|

the

| Page 8: [56] Deleted | Natalie Ceperley | 28.06.17 11:22 |
|---|---|---|

has

| Page 8: [56] Deleted | Natalie Ceperley | 28.06.17 11:22 |
|---|---|---|

has

| Page 8: [57] Deleted | Natalie Ceperley | 08.06.17 15:07 |
|---|---|---|

timing

| Page 8: [57] Deleted | Natalie Ceperley | 08.06.17 15:07 |
|---|---|---|

timing

| Page 8: [57] Deleted | Natalie Ceperley | 08.06.17 15:07 |
|---|---|---|

timing

| Page 8: [57] Deleted | Natalie Ceperley | 08.06.17 15:07 |
|---|---|---|

timing

| Page 8: [57] Deleted | Natalie Ceperley | 08.06.17 15:07 |
|---|---|---|

timing

| Page 8: [57] Deleted | Natalie Ceperley | 08.06.17 15:07 |
|---|---|---|

timing

| Page 8: [57] Deleted | Natalie Ceperley | 08.06.17 15:07 |
|---|---|---|

timing

| Page 8: [57] Deleted | Natalie Ceperley | 08.06.17 15:07 |
|---|---|---|

timing

| Page 8: [57] Deleted | Natalie Ceperley | 08.06.17 15:07 |

timing

| Page 8: [58] Deleted | Natalie Ceperley | 04.07.17 00:58 |

There is

| Page 8: [58] Deleted | Natalie Ceperley | 04.07.17 00:58 |

There is

| Page 8: [58] Deleted | Natalie Ceperley | 04.07.17 00:58 |

There is

| Page 8: [58] Deleted | Natalie Ceperley | 04.07.17 00:58 |

There is

| Page 8: [58] Deleted | Natalie Ceperley | 04.07.17 00:58 |

There is

| Page 8: [58] Deleted | Natalie Ceperley | 04.07.17 00:58 |

There is

| Page 8: [58] Deleted | Natalie Ceperley | 04.07.17 00:58 |

There is

| Page 9: [59] Deleted | Microsoft Office User | 01.02.17 17:15 |

There is a scale discrepancy between the eddy covariance measurements and the net radiometer measurements since the latter only senses exchanges directly above and below it whereas the former's range of detection can span a larger area depending on the wind speed. To account for this, we modeled the net radiation at each small station and then compared it to that measured with net radiometers with acceptable results (figure 3).

| Page 9: [60] Deleted | Natalie Ceperley | 02.02.17 20:09 |

Further examination reveals that surface temperature and net long wave radiation have a strong negative correlation (figure 7see supplementary material) that is controlled by variations in soil moisture.

| Page 12: [61] Deleted | Natalie Ceperley | 04.07.17 01:56 |

prove

| Page 12: [61] Deleted | Natalie Ceperley | 04.07.17 01:56 |

prove

| Page 12: [61] Deleted | Natalie Ceperley | 04.07.17 01:56 |

prove

| Page 12: [61] Deleted | Natalie Ceperley | 04.07.17 01:56 |

prove

[revised manuscript text omitted]

Space After:  0 pt

| Page 23: [66] Formatted Table | Natalie | 26.06.17 16:20 |
| --- | --- | --- |

Formatted Table

| Page 23: [67] Formatted | Natalie | 26.06.17 16:17 |
| --- | --- | --- |

Font:Not Bold, Italic

| Page 23: [68] Formatted | Natalie | 26.06.17 16:17 |
| --- | --- | --- |

Space After:  0 pt

| Page 23: [69] Formatted | Natalie | 26.06.17 16:17 |
| --- | --- | --- |

Font:Not Bold

| Page 23: [70] Formatted | Natalie | 26.06.17 16:17 |
| --- | --- | --- |

Font:Not Bold, Italic

| Page 23: [71] Formatted | Natalie | 26.06.17 16:17 |
| --- | --- | --- |

Space After:  0 pt

| Page 23: [72] Formatted | Natalie | 26.06.17 16:17 |
| --- | --- | --- |

Font:Not Bold

| Page 23: [73] Formatted | Natalie | 26.06.17 16:17 |
|---|---|---|

Font:Not Bold, Italic

| Page 23: [74] Formatted | Natalie | 26.06.17 16:17 |
|---|---|---|

Space After:  0 pt

| Page 23: [75] Formatted | Natalie | 26.06.17 16:17 |
|---|---|---|

Font:Not Bold

| Page 23: [76] Formatted | Natalie | 26.06.17 16:17 |
|---|---|---|

Font:Not Bold, Italic

| Page 23: [77] Formatted | Natalie | 26.06.17 16:17 |
|---|---|---|

Space After:  0 pt

| Page 23: [78] Formatted | Natalie | 26.06.17 16:17 |
|---|---|---|

Font:Not Bold

| Page 23: [79] Formatted | Natalie | 26.06.17 16:17 |
|---|---|---|

Font:Not Bold, Italic

| Page 23: [80] Formatted | Natalie | 26.06.17 16:17 |
|---|---|---|

Space After:  0 pt

| Page 23: [81] Formatted | Natalie | 26.06.17 16:17 |
|---|---|---|

Font:Not Bold

| Page 23: [82] Formatted | Natalie | 26.06.17 16:17 |
|---|---|---|

Font:Not Bold, Italic

| Page 23: [83] Formatted | Natalie | 26.06.17 16:17 |
|---|---|---|

Space After:  0 pt

| Page 23: [84] Formatted | Natalie | 26.06.17 16:17 |
|---|---|---|

Font:Not Bold

| Page 23: [85] Formatted | Natalie | 26.06.17 16:17 |
|---|---|---|

Font:Not Bold, Italic

| Page 23: [86] Formatted | Natalie | 26.06.17 16:17 |
|---|---|---|

Space After:  0 pt

| Page 23: [87] Formatted | Natalie | 26.06.17 16:17 |
|---|---|---|

Font:Not Bold

| Page 23: [88] Formatted | Natalie | 26.06.17 16:17 |
|---|---|---|

Font:Not Bold, Italic

| | | |
|---|---|---|
| **Page 23: [89] Formatted** | **Natalie** | **26.06.17 16:17** |

Space After:  0 pt

| | | |
|---|---|---|
| **Page 23: [90] Formatted** | **Natalie** | **26.06.17 16:17** |

Font:Not Bold

| | | |
|---|---|---|
| **Page 23: [91] Formatted** | **Natalie** | **26.06.17 16:17** |

Font:Not Bold, Italic

| | | |
|---|---|---|
| **Page 23: [92] Formatted** | **Natalie** | **26.06.17 16:17** |

Space After:  0 pt

| | | |
|---|---|---|
| **Page 23: [93] Formatted** | **Natalie** | **26.06.17 16:17** |

Font:Not Bold

| | | |
|---|---|---|
| **Page 23: [94] Formatted** | **Natalie** | **26.06.17 16:17** |

Font:Not Bold, Italic

| | | |
|---|---|---|
| **Page 23: [95] Formatted** | **Natalie** | **26.06.17 16:17** |

Space After:  0 pt

| | | |
|---|---|---|
| **Page 23: [96] Formatted** | **Natalie** | **26.06.17 16:17** |

Font:Not Bold

| | | |
|---|---|---|
| **Page 23: [97] Formatted** | **Natalie** | **26.06.17 16:17** |

Font:Not Bold, Italic

| | | |
|---|---|---|
| **Page 23: [98] Formatted** | **Natalie** | **26.06.17 16:17** |

Space After:  0 pt

| | | |
|---|---|---|
| **Page 23: [99] Formatted** | **Natalie** | **26.06.17 16:17** |

Font:Not Bold

| | | |
|---|---|---|
| **Page 23: [100] Formatted** | **Natalie** | **26.06.17 16:17** |

Font:Not Bold, Italic

| | | |
|---|---|---|
| **Page 23: [101] Formatted** | **Natalie** | **26.06.17 16:17** |

Space After:  0 pt

| | | |
|---|---|---|
| **Page 23: [102] Formatted** | **Natalie** | **26.06.17 16:17** |

Font:Not Bold

| | | |
|---|---|---|
| **Page 23: [103] Formatted** | **Natalie** | **26.06.17 16:17** |

Font:Not Bold, Italic

| | | |
|---|---|---|
| **Page 23: [104] Formatted** | **Natalie** | **26.06.17 16:17** |

Space After:  0 pt

| Page 23: [105] Formatted | Natalie | 26.06.17 16:17 |
|---|---|---|

Font:Not Bold

| Page 23: [106] Formatted | Natalie | 26.06.17 16:17 |
|---|---|---|

Font:Not Bold, Italic

| Page 23: [107] Formatted | Natalie | 26.06.17 16:17 |
|---|---|---|

Space After:  0 pt

| Page 23: [108] Formatted | Natalie | 26.06.17 16:17 |
|---|---|---|

Font:Not Bold

| Page 23: [109] Formatted | Natalie | 26.06.17 16:17 |
|---|---|---|

Font:Not Bold, Italic

| Page 23: [110] Formatted | Natalie | 26.06.17 16:17 |
|---|---|---|

Space After:  0 pt

| Page 23: [111] Formatted | Natalie | 26.06.17 16:17 |
|---|---|---|

Font:Not Bold

| Page 23: [112] Formatted | Natalie | 26.06.17 16:17 |
|---|---|---|

Font:Not Bold, Italic

| Page 23: [113] Formatted | Natalie | 26.06.17 16:17 |
|---|---|---|

Space After:  0 pt

| Page 23: [114] Formatted | Natalie | 26.06.17 16:17 |
|---|---|---|

Font:Not Bold

| Page 23: [115] Formatted | Natalie | 26.06.17 16:17 |
|---|---|---|

Font:Not Bold, Italic

| Page 23: [116] Formatted | Natalie | 26.06.17 16:17 |
|---|---|---|

Space After:  0 pt

| Page 23: [117] Formatted | Natalie | 26.06.17 16:17 |
|---|---|---|

Font:Not Bold

| Page 23: [118] Formatted | Natalie | 26.06.17 16:17 |
|---|---|---|

Font:Not Bold, Italic

| Page 23: [119] Formatted | Natalie | 26.06.17 16:17 |
|---|---|---|

Space After:  0 pt

| Page 23: [120] Formatted | Natalie | 26.06.17 16:17 |
|---|---|---|

Font:Not Bold

| Page 23: [121] Formatted | Natalie | 26.06.17 16:17 |
|---|---|---|

Font:Not Bold, Italic

| Page 23: [122] Formatted | Natalie | 26.06.17 16:17 |
|---|---|---|

Space After:  0 pt

| Page 23: [123] Formatted | Natalie | 26.06.17 16:17 |
|---|---|---|

Font:Not Bold

| Page 23: [124] Formatted | Natalie | 26.06.17 16:17 |
|---|---|---|

Font:Not Bold, Italic

| Page 23: [125] Formatted | Natalie | 26.06.17 16:17 |
|---|---|---|

Space After:  0 pt

| Page 23: [126] Formatted | Natalie | 26.06.17 16:17 |
|---|---|---|

Font:Not Bold

| Page 23: [127] Formatted | Natalie | 26.06.17 16:17 |
|---|---|---|

Font:Not Bold, Italic

| Page 23: [128] Formatted | Natalie | 26.06.17 16:17 |
|---|---|---|

Space After:  0 pt

| Page 23: [129] Formatted | Natalie | 26.06.17 16:17 |
|---|---|---|

Font:Not Bold

| Page 23: [130] Formatted | Natalie | 26.06.17 16:17 |
|---|---|---|

Font:Not Bold, Italic

| Page 23: [131] Formatted | Natalie | 26.06.17 16:17 |
|---|---|---|

Space After:  0 pt

| Page 23: [132] Formatted | Natalie | 26.06.17 16:17 |
|---|---|---|

Font:Not Bold

| Page 23: [133] Formatted | Natalie | 26.06.17 16:17 |
|---|---|---|

Font:Not Bold, Italic

| Page 23: [134] Formatted | Natalie | 26.06.17 16:17 |
|---|---|---|

Space After:  0 pt

| Page 23: [135] Formatted | Natalie | 26.06.17 16:17 |
|---|---|---|

Font:Not Bold

| Page 23: [136] Formatted | Natalie | 26.06.17 16:17 |
|---|---|---|

Font:Not Bold

| Page 23: [137] Formatted | Natalie | 26.06.17 16:17 |
|---|---|---|

Space After: 0 pt, No bullets or numbering, Tabs:Not at 8 cm + 16 cm

| Page 23: [138] Formatted | Natalie | 26.06.17 16:17 |
|---|---|---|

Space After: 0 pt

| Page 23: [139] Formatted | Natalie | 26.06.17 16:17 |
|---|---|---|

Font:Not Bold

| Page 23: [140] Formatted | Natalie | 26.06.17 16:17 |
|---|---|---|

Space After: 0 pt

| Page 23: [141] Formatted | Natalie | 26.06.17 16:17 |
|---|---|---|

Font:Not Bold

| Page 23: [142] Formatted | Natalie | 26.06.17 16:17 |
|---|---|---|

Space After: 0 pt

| Page 23: [143] Formatted | Natalie | 26.06.17 16:17 |
|---|---|---|

Font:Not Bold

| Page 23: [144] Formatted | Natalie | 26.06.17 16:17 |
|---|---|---|

Space After: 0 pt

| Page 23: [145] Formatted | Natalie | 26.06.17 16:17 |
|---|---|---|

Font:Not Bold

| Page 23: [146] Formatted | Natalie | 26.06.17 16:17 |
|---|---|---|

Space After: 0 pt

| Page 23: [147] Formatted | Natalie | 26.06.17 16:17 |
|---|---|---|

Font:Not Bold

| Page 23: [148] Formatted | Natalie | 26.06.17 16:17 |
|---|---|---|

Space After: 0 pt

| Page 23: [149] Formatted | Natalie | 26.06.17 16:17 |
|---|---|---|

Font:Not Bold

| Page 23: [150] Formatted | Natalie | 26.06.17 16:17 |
|---|---|---|

Space After: 0 pt

| Page 23: [151] Formatted | Natalie | 26.06.17 16:17 |
|---|---|---|

Font:Not Bold

| Page 23: [152] Formatted | Natalie | 26.06.17 16:17 |
|---|---|---|

Space After: 0 pt

| Page 23: [153] Formatted | Natalie | 26.06.17 16:17 |

Font:Not Bold

| Page 23: [154] Formatted | Natalie | 26.06.17 16:17 |

Space After:  0 pt

| Page 23: [155] Formatted | Natalie | 26.06.17 16:17 |

Font:Not Bold

| Page 23: [156] Formatted | Natalie | 26.06.17 16:17 |

Space After:  0 pt

| Page 23: [157] Formatted | Natalie | 26.06.17 16:17 |

Font:Not Bold

| Page 23: [158] Formatted | Natalie | 26.06.17 16:17 |

Space After:  0 pt

| Page 23: [159] Formatted | Natalie | 26.06.17 16:17 |

Font:Not Bold

| Page 23: [160] Formatted | Natalie | 26.06.17 16:17 |

Space After:  0 pt

| Page 23: [161] Formatted | Natalie | 26.06.17 16:17 |

Font:Not Bold

| Page 23: [162] Formatted | Natalie | 26.06.17 16:17 |

Space After:  0 pt,  No bullets or numbering, Tabs:Not at  8 cm +  16 cm

| Page 23: [163] Formatted | Natalie | 26.06.17 16:17 |

Space After:  0 pt

| Page 23: [164] Formatted | Natalie | 26.06.17 16:17 |

Font:Not Bold

| Page 23: [165] Formatted | Natalie | 26.06.17 16:17 |

Space After:  0 pt

| Page 23: [166] Formatted | Natalie | 26.06.17 16:17 |

Font:Not Bold

| Page 23: [167] Formatted | Natalie | 26.06.17 16:17 |

Space After:  0 pt

| Page 23: [168] Formatted | Natalie | 26.06.17 16:17 |

Font:Not Bold

| Page 23: [169] Formatted | Natalie | 26.06.17 16:17 |

Space After: 0 pt

| Page 23: [170] Formatted | Natalie | 26.06.17 16:17 |

Font:Not Bold

| Page 23: [171] Formatted | Natalie | 26.06.17 16:17 |

Space After: 0 pt

| Page 23: [172] Formatted | Natalie | 26.06.17 16:17 |

Font:Not Bold

| Page 23: [173] Formatted | Natalie | 26.06.17 16:17 |

Space After: 0 pt

| Page 24: [174] Formatted | Natalie | 26.06.17 16:17 |

Font:Not Bold

| Page 24: [175] Formatted | Natalie | 26.06.17 16:17 |

Space After: 0 pt

| Page 24: [176] Formatted | Natalie | 26.06.17 16:17 |

Font:Not Bold

| Page 24: [177] Formatted | Natalie | 26.06.17 16:17 |

Space After: 0 pt

| Page 24: [178] Formatted | Natalie | 26.06.17 16:17 |

Font:Not Bold

| Page 24: [179] Formatted | Natalie | 26.06.17 16:17 |

Space After: 0 pt

| Page 24: [180] Formatted | Natalie | 26.06.17 16:17 |

Font:Not Bold

| Page 24: [181] Formatted | Natalie | 26.06.17 16:17 |

Space After: 0 pt, No bullets or numbering, Tabs:Not at 8 cm + 16 cm

| Page 24: [182] Formatted | Natalie | 26.06.17 16:17 |

Space After: 0 pt

| Page 24: [183] Formatted | Natalie | 26.06.17 16:17 |

Font:Not Bold

| Page 24: [184] Formatted | Natalie | 26.06.17 16:17 |

Space After: 0 pt

| Page 24: [185] Formatted | Natalie | 26.06.17 16:17 |

Font:Not Bold

| Page 24: [186] Formatted | Natalie | 26.06.17 16:17 |

Space After:  0 pt

| Page 24: [187] Formatted | Natalie | 26.06.17 16:17 |

Space After:  0 pt,  No bullets or numbering, Tabs:Not at  8 cm +  16 cm

| Page 24: [188] Formatted | Natalie | 26.06.17 16:17 |

Space After:  0 pt

| Page 24: [189] Formatted | Natalie | 26.06.17 16:17 |

Font:Not Bold

| Page 24: [190] Formatted | Natalie | 26.06.17 16:17 |

Space After:  0 pt

| Page 24: [191] Formatted | Natalie | 26.06.17 16:17 |

Font:Not Bold

| Page 24: [192] Formatted | Natalie | 26.06.17 16:17 |

Space After:  0 pt,  No bullets or numbering, Tabs:Not at  8 cm +  16 cm

| Page 24: [193] Formatted | Natalie | 26.06.17 16:17 |

Space After:  0 pt

| Page 24: [194] Formatted | Natalie | 26.06.17 16:17 |

Font:Not Bold

| Page 24: [195] Formatted | Natalie | 26.06.17 16:17 |

Space After:  0 pt

| Page 24: [196] Formatted | Natalie | 26.06.17 16:17 |

Font:Not Bold

| Page 24: [197] Formatted | Natalie | 26.06.17 16:17 |

Space After:  0 pt

| Page 24: [198] Formatted | Natalie | 26.06.17 16:17 |

Font:Not Bold

| Page 24: [199] Formatted | Natalie | 26.06.17 16:17 |

Space After:  0 pt

| Page 24: [200] Formatted | Natalie | 26.06.17 16:17 |

Font:Not Bold

| Page 24: [201] Formatted | Natalie | 26.06.17 16:17 |
| --- | --- | --- |

Space After:  0 pt

| Page 24: [202] Formatted | Natalie | 26.06.17 16:17 |
| --- | --- | --- |

Space After:  0 pt,  No bullets or numbering, Tabs:Not at  8 cm +  16 cm

| Page 24: [203] Formatted | Natalie | 26.06.17 16:17 |
| --- | --- | --- |

Space After:  0 pt

| Page 24: [204] Formatted | Natalie | 26.06.17 16:17 |
| --- | --- | --- |

Font:Not Bold

| Page 24: [205] Formatted | Natalie | 26.06.17 16:17 |
| --- | --- | --- |

Space After:  0 pt

| Page 24: [206] Deleted | Natalie Ceperley | 28.06.17 14:41 |
| --- | --- | --- |

 Classification based on Chorology and Life Form is based on Raunkiaer (1934), Schnell (1971), and Keay & Hepper (1954-1972) via Adomou (2005).

| Page 24: [207] Deleted | Microsoft Office User | 01.02.17 13:00 |
| --- | --- | --- |

[Figure]

**Figure 2: Distribution of albedo by landcover (Agricultural, Wetland, and Forest) and by month for the study period. The central dot is the median, the edges of the box are the 25th and 75th percentiles, the whiskers extend to the most extreme data points not considered outliers, and outliers are plotted individually. Average values for various land-covers listed by Brutsaert (1982) are drawn and labelled to the right. Ranges for these are shown in**

**dotted lines. Our values mostly range between those of bare field or green grass and dry soils and desert, with dry prairie or savanna making a good mid value.**

[Figure]

|  | Agriculture | Savanna-Forest |
|---|---|---|
| a) $LW_{net}$ | 0.58 | 0.50 |
| b) $SW_{net}$ | 0.76 | 0.78 |
| c) $R_n$ | 0.80 | 0.77 |
| $R_n$-H- $L_eE$ | 0.86 | 0.86 |

**Figure 3: Comparison of measurements and calculations on a half hour scale of (a) net short wave, (b) net long wave, and (c) total radiation and the residual (not shown, correlation in table) after subtracting the measured turbulent fluxes. In all cases the slope is less than 1, which means that the measurements are greater than the calculation. The slopes of the regression lines are in the accompanying table. In each plot, measurements over the agricultural field are shown in blue and over the savanna-forest are shown in green. Black lines show the one to one line and coloured lines show the least squared regression lines.**

| Page 24: [208] Formatted | Unknown | |
|---|---|---|

Font:Times

| Page 29: [209] Deleted | Natalie Ceperley | 02.02.17 19:43 |
|---|---|---|

[Figure]

**Figure 7: Net Long-wave Radiation Response to Soil Moisture and Surface Temperature.** Surface temperature and net long wave radiation had a strong negative correlation with each other and also with volumetric soil water content, shown by the colour. The pattern of this correlation varied by land cover, savanna-forest above and agriculture below.

**Page 33: [210] Moved to page 32 (Move #2) Natalie Ceperley**            **02.02.17 19:45**

[Figure]

**Page 33: [211] Moved to page 33 (Move #1) Natalie Ceperley**          **02.02.17 19:45**

Time series of evaporative fraction. Upper plot shows evaporative fraction over the field and lower plot shows evaporative fraction over the savanna-forest. Measured data points are in black (points), calculated evaporative fraction based on soil moisture and NDVI is shown in blue and just NDVI is shown in green.